# Osmolarity-independent electrical cues guide rapid response to injury in zebrafish epidermis

Andrew S Kennard[1,2], Julie A Theriot[2]*

[1]Biophysics Program, Stanford University, Stanford, United States; [2]Department of Biology and Howard Hughes Medical Institute, University of Washington, Seattle, United States

**Abstract** The ability of epithelial tissues to heal after injury is essential for animal life, yet the mechanisms by which epithelial cells sense tissue damage are incompletely understood. In aquatic organisms such as zebrafish, osmotic shock following injury is believed to be an early and potent activator of a wound response. We find that, in addition to sensing osmolarity, basal skin cells in zebrafish larvae are also sensitive to changes in the particular ionic composition of their surroundings after wounding, specifically the concentration of sodium chloride in the immediate vicinity of the wound. This sodium chloride-specific wound detection mechanism is independent of cell swelling, and instead is suggestive of a mechanism by which cells sense changes in the transepithelial electrical potential generated by the transport of sodium and chloride ions across the skin. Consistent with this hypothesis, we show that electric fields directly applied within the skin are sufficient to initiate actin polarization and migration of basal cells in their native epithelial context in vivo, even overriding endogenous wound signaling. This suggests that, in order to mount a robust wound response, skin cells respond to both osmotic and electrical perturbations arising from tissue injury.

*For correspondence:
jtheriot@uw.edu

**Competing interests:** The authors declare that no competing interests exist.

## Introduction

Epithelial tissues separate organisms from the outside world and bear an ever-present risk of injury, and so epithelial wound healing is a critical homeostatic process. The initial wound response in many tissues consists of rearrangements of the actomyosin cytoskeleton to form a contractile purse string that closes the wound in concert with protrusive actin structures that promote cell migration toward the injury and help cover the damaged area (*Abreu-Blanco et al., 2012*; *Eming et al., 2014*; *Rothenberg and Fernandez-Gonzalez, 2019*). In order to mount a wound response, cells must detect both the presence of a wound and the direction in which the wound is located; this information must be ultimately derived from changes in the cell's environment caused by injury. Much progress has been made in understanding how cells coordinate a wound response locally using mechanical cues, such as the presence of free space adjacent to a cell, or forces exerted by neighboring cells via cell–cell junctions (*Begnaud et al., 2016*). Yet many internal signaling events—including waves of calcium influx, hydrogen peroxide release, and purinergic signaling—occur in cells hundreds of microns away from a wound less than a minute after injury, suggesting that there must be other mechanisms by which cells sense injury on a tissue-wide scale (*Niethammer et al., 2009*; *Razzell et al., 2013*; *Xu and Chisholm, 2011*; *Yin et al., 2007*; *Yoo et al., 2012*). This large-scale coordination could arise from rapid changes in the external environment around each cell, which is altered by the breach of the epithelial barrier and intermixing between formerly separated compartments. Such environmental changes have been challenging to characterize.

Non-keratinized epithelia exposed to aqueous environments—such as the mucosal surfaces of the human body or the skin of aquatic organisms—devote considerable energy to regulate ion transport between the environment and internal body fluids. This ion transport maintains the distinct composition and osmolarity of interstitial fluid relative to the surrounding environment: for example, interstitial fluid in saltwater fishes is hyposmotic relative to the environment, while in freshwater fishes it is hyperosmotic (*Boisen et al., 2003*; *Potts, 1984*). The driving force for ion transport derives from electrically active energy-requiring pumps such as the sodium–potassium ATPase and the V-ATPase proton pump, which establish electrochemical gradients that are coupled to the transport of other ionic species through passive ion channels and transporters. Differential transport of ions can lead to a net electric charge on one side of the epithelium relative to the other, which creates a voltage gradient between the two sides of the epithelial layer known as the transepithelial potential (TEP). This TEP is related to but distinct from the transmembrane potential between the inside and outside of each cell, and it is sensitive to the composition of the solutions on either side of the epithelium (*Dietz et al., 1967*; *Ussing and Zerahn, 1951*). Ion substitution and pharmacological inhibition of ion channels suggests that transport of sodium and chloride ions is essential for maintaining the TEP in different tissues (*Dietz et al., 1967*; *McCaig and Robinson, 1982*; *Reid et al., 2005*). If the tissue is damaged, fluid intermixing disrupts the normal ion gradients and leads to a variety of osmotic and electrical changes in the environment of the epithelium, including osmotic shock and short-circuiting of the TEP.

Both osmotic and electrical changes could act as early cues of injury acting at a tissue-wide scale. In *Xenopus laevis* (clawed frog) and *Danio rerio* (zebrafish) larvae, the wound response is inhibited when the composition of the external medium resembles that of interstitial fluid (*Fuchigami et al., 2011*; *Gault et al., 2014*), but this observation alone cannot distinguish between osmotic and electrical mechanisms. Crucially, the osmotic and electrical mechanisms for sensing tissue damage are physically intertwined, and it is unclear how each signal distinctly contributes to the wound response in aqueous environments.

Regarding osmotic cues, in zebrafish epidermal cells, cell swelling due to osmotic shock following injury has been shown to provide a physical, cell-autonomous cue of tissue damage, and this cue is amplified and relayed to other cells with subsequent extracellular ATP release (*Gault et al., 2014*). In addition to promoting signaling at the tissue level, osmotic swelling could also mechanically promote migration at the cellular level: hypotonic shock can promote formation of lamellipodia (*Chen et al., 2019*) and can intrinsically stabilize a polarized actin cytoskeleton by increasing mechanical feedback through membrane tension (*Houk et al., 2012*).

A major focus of previous investigations into electrical activity in vivo is the consequence of small electric currents that emanate from tissue for hours and days during development and regeneration (*Ferreira et al., 2016*; *Rajnicek et al., 1988*; *Robinson, 1983*; *Tseng et al., 2010*). Less is known about the possible role of electric fields in guiding cell migration in the early phase of wound healing, within the first few minutes or hours after injury. Electric currents have been directly measured emanating from wounds in many animal tissues in this early phase, including adult zebrafish skin, rat cornea and skin, tails of newt and *Xenopus* tadpoles, bronchial epithelia of rhesus macaques, and even human skin (*Ferreira et al., 2016*; *Huang et al., 2009*; *Li et al., 2012*; *Nawata, 2001*; *Reid et al., 2009*; *Reid et al., 2007*; *Reid et al., 2005*; *Sun et al., 2011*). The currents measured emanating from these wounds are ~10–100 times stronger than regeneration or developmental currents in the same model systems (*Ferreira et al., 2016*; *Reid et al., 2005*; *Robinson, 1983*). In rat cornea, pharmacological perturbations that increase or decrease the magnitude of the wound current also correspondingly increase or decrease the rate of wound closure, suggesting that electrical currents may aid in healing (*Reid et al., 2005*). However, the effect of electrical currents on wound healing in vivo has only been measured at a coarse-grained scale, and it is unclear how electrical fields in vivo affect subcellular dynamics of individual epithelial cells. Furthermore, only a few attempts have been made to apply exogenous electric fields through tissues in living animals to determine directly how electric fields alter cell behavior in vivo, and only on time scales longer than an hour (*Borgens et al., 1977*; *Chiang et al., 1991*; *Hotary and Robinson, 1994*).

The response of cultured cells to applied electric fields has been better studied than responses in vivo, and it has been observed that a wide variety of cell types migrate directionally in the presence of an electric field (*Allen et al., 2013*; *Riding and Pullar, 2016*; *Sun et al., 2011*). Importantly, most cells appear to be responsive not to the magnitude of the electric field per se (in V/cm), but rather

to the current density in their surroundings (in mA/cm$^2$), which is directly proportional to the electric field, scaled by the conductivity of the medium (*Allen et al., 2013*). The signaling networks that allow cells to respond to electric fields are still being unraveled, but a prevailing model is that electric fields drag charged membrane proteins to one side of the cell by electrophoresis, with the resulting asymmetric protein distribution leading to cytoskeletal polarization, mediated by key signal transduction factors including phosphoinositide-3-kinase (PI3K) (*Allen et al., 2013*; *Sarkar et al., 2019*; *Zhao et al., 2006*). The response of cells to electric fields is highly reproducible, allowing precise control of cell movement in culture using only electrical cues (*Cohen et al., 2014*; *Zajdel et al., 2020*). Intriguingly, cells respond almost immediately (within seconds or minutes) to electric fields applied in culture, raising the possibility that they could also be responsive to the electric fields generated immediately after wounding in vivo. However, a tractable system in which to observe the rapid response of cells to electric fields in vivo has been lacking.

Due to their optical transparency and ease of experimental manipulation, zebrafish larvae have been an important model system for understanding the rapid sequence of events following tissue damage, in particular the response of epidermal cells to injury in the tailfin (*Franco et al., 2019*; *Gault et al., 2014*; *Mateus et al., 2012*; *Yoo et al., 2012*). The zebrafish larval epidermis is bilayered, with a superficial cell layer originating from the enveloping layer in early embryogenesis, and a basal cell layer that resides on a collagenous basal lamina and is specified by the $\Delta Np63$ promoter (*Bakkers et al., 2002*; *Le Guellec et al., 2004*; *Rasmussen et al., 2015*; *Sonawane et al., 2005*). Because zebrafish are freshwater organisms, the osmotic gradient across the zebrafish epidermis is large: external culture medium has an osmolarity of about 10 mOsmol/l while the osmolarity of interstitial fluid inside the fish is estimated to be about 270–300 mOsmol/l (*Boisen et al., 2003*; *Gault et al., 2014*). This gradient is maintained by a variety of specialized ion-transporting cells known as ionocytes that span across the two epidermal cell layers (*Guh et al., 2015*).

Previous work in zebrafish has shown that, within seconds after injury, the basal cell layer reacts to tissue damage primarily by active cell migration while the superficial layer reacts by purse string contraction around the wound (*Gault et al., 2014*). The speed of the wound response in this tissue implies that equally rapid environmental changes must initiate this process. Although osmotic changes have been identified as one wound response cue (*Gault et al., 2014*), osmotic changes alone lack directional information and can only signal to a cell that a wound has occurred, and possibly the distance between that cell and the wound; osmotic cues must be combined with other cues to determine the *direction* of the wound in relation to any individual cell. Electrical perturbations accompanying a drop in external osmotic pressure would in principle provide a natural directional cue to guide cell polarization and migration in the wound response, but the role of electric fields in guiding migration in vivo in the first few minutes after injury has not been explored.

Here, we focus on the cues that specifically initiate the cell migration behaviors associated with the early wound response in the zebrafish epidermis. Through live imaging of the actin cytoskeleton in basal cells immediately following injury, we observe a range of differences in the initial wound response under different environmental conditions. We report that, in addition to the 'osmotic surveillance' mechanism for wound detection previously identified, zebrafish epidermal cells are also sensitive to ion-specific cues following tissue damage, independent of osmolarity and cell swelling. These ion-specific cues are consistent with expected changes in electrical activity at wounds, and in support of this mechanism, we show that electric fields are capable of guiding cells and overriding endogenous wound cues, suggesting that disruption of the electrical properties of tissues may be an important injury signal in the zebrafish epidermis.

## Results

### Tissue laceration induces a rapid and coordinated wound response

A variety of wounding techniques have been used to observe the injury response in the zebrafish tailfin, including tail transection with a scalpel, laser wounding, and burn wounding (*Gault et al., 2014*; *Miskolci et al., 2019*; *Yoo et al., 2012*). We were specifically interested in the migratory response immediately following tissue damage, and so we developed a wounding technique—which we refer to as tissue laceration—which led to a strong and reproducible early migratory response to injury. In our laceration approach, a glass rod is pulled to a fine point, and the tissue is impaled with

this needle at locations dorsal and ventral to the terminus of the notochord. The needle is dragged in a posterior direction through the surrounding tissue, tearing the tailfin (*Figure 1A*).

We found that lacerated tissue rapidly reorganized and contracted around the wound site over a period of about 15–20 min (*Figure 1B*), consistent with wound closure observed with other methods mentioned above. A direct comparison with tail transection revealed similar spatial patterns of tissue rearrangement (*Figure 1—figure supplement 1A*). In timelapse videos of wounds from both techniques, laceration wounds induce a more pronounced migratory response within the first few minutes after wounding, suggesting that laceration wounds may be ideal for studying the early stages of the wound response (*Figure 1—video 1*). To determine whether this tissue reorganization was mediated at least in part by actin-based migration of cells in the basal layer, we investigated actin organization during wound closure using a basal cell-specific Gal4 driver fish crossed to a fish expressing LifeAct-EGFP from the UAS promoter.

Inspection of wound closure at high magnification revealed dynamic cytoskeletal rearrangements accompanying basal cell migration (*Figure 1C–D*; *Figure 1—videos 2* and *3*). Prior to wounding, the basal cell F-actin distribution was enriched uniformly around the cell at cell–cell junctions (*Figure 1B*, first image). Within two minutes of injury, actin polarized with significant accumulation of LifeAct-labeled filamentous structures on the wound-facing edges of the cell, and the formerly static cell boundary began to rapidly protrude and retract on a sub-minute timescale, reminiscent of actin ruffling (*Figure 1C*; *Figure 1—video 2*). For cells close to the wound, these dynamic actin-rich ruffles stabilized into lamellipodial sheets that protruded rapidly, causing the cells to elongate and translocate parallel to their long axis. This differs from the behavior of these cells when isolated in culture, where they have been studied extensively for their rapid and persistent migration and are often referred to as keratocytes (*Lou et al., 2015*). Isolated migratory basal epidermal cells typically adopt a wider, 'canoe-shaped' morphology and move perpendicular to their long axis with protrusions that maintain a persistent shape during migration (*Keren et al., 2008*). In vivo, the polarized actin ruffling response and cell elongation toward the direction of the wound was apparent for basal cells up to several hundred micrometers away, although these distant cells typically did not physically translocate (*Figure 1—video 3*). Following this initial rapidly dynamic wound response, the basal cells retracted their protrusive actin structures and their shapes gradually returned to resemble those of cells in an unwounded larva, though for roughly 30 min post wounding the cell–cell junctions continued to protrude and retract on a small scale. The migratory phase of the wound response process was rapid, with cells polarizing, migrating, and stopping within about 15–20 min after injury.

To better compare complex migratory behavior among many larvae and across distinct experimental conditions, we measured the local speed of cells in the basal cell monolayer over time. To do this, we developed a velocimetry analysis pipeline based on tracking the movement of many computationally detected feature points (see *Figure 1—figure supplement 1C* and *Methods*). The speed of these feature points was locally averaged in space and time to reveal the coordinated wave-like propagation of cell speed originating at the wound and traveling away (*Figure 1E*), consistent with studies of other wound types (*Gault et al., 2014*). Interestingly, laceration prompted a stronger cell migration response within the first 5 min of injury compared with tail transection (*Figure 1—figure supplement 1B*)—perhaps due to mechanical differences between the two methods, such as the increased tissue stretch applied with the laceration wound—and the profile of cell speed in space and time was reproducible across larvae despite the variation in the shapes of the laceration-induced wound margins as compared to other wounding protocols (*Figure 1E*, line graphs).

We wondered if laceration might induce transient increases in cytoplasmic calcium concentration, which have been observed with other wounding techniques (*Antunes et al., 2013*; *Enyedi et al., 2016*; *Razzell et al., 2013*; *Xu and Chisholm, 2011*; *Yoo et al., 2012*). To test this we injected embryos carrying the basal cell Gal4 driver with a *UAS:GCaMP6f-P2A-nls-dTomato* plasmid to express the calcium indicator GCaMP6f and a nuclear-localized dTomato fluorescent protein mosaically in basal cells. Consistent with observations from other model systems and wounding methods, we found intense and rapid propagation of increased calcium levels throughout the tailfin (*Figure 1F*; *Figure 1—video 4*). This calcium wave propagated about five times faster than the cell migration wave (200 µm/min vs. 40 µm/min, *Figure 1G*), implying that there is no uniform time delay between the calcium transient in a cell and the onset of migration. The same trend is observed when directly comparing the average changes in cell migratory speed and in GCaMP6f fluorescence intensity over time, normalized to comparable dimensionless quantities, where the change in GCaMP6f

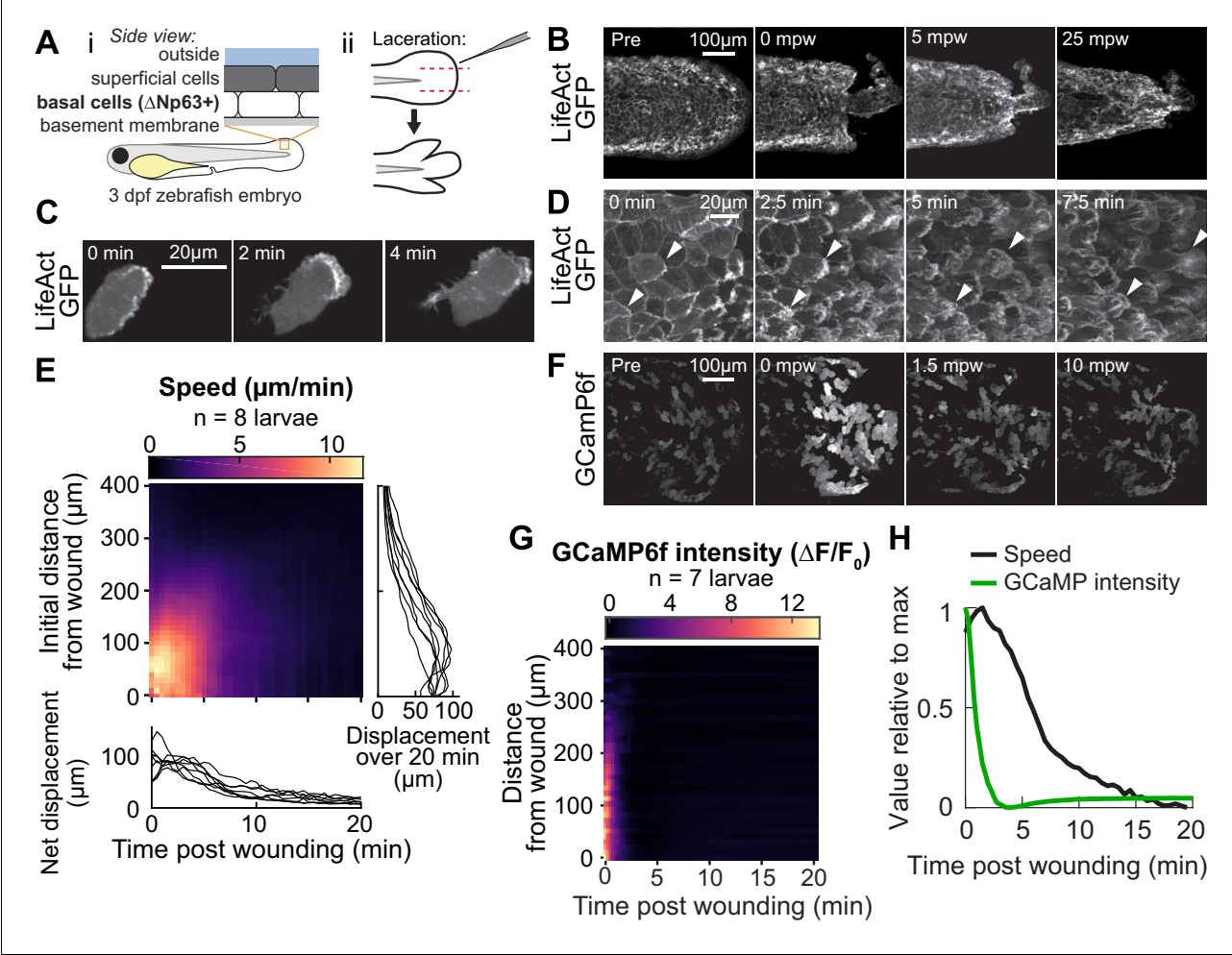

**Figure 1.** Tissue laceration induces a rapid and coordinated wound response. (A) Schematic of (*i*) bilayered larval zebrafish skin and (*ii*) laceration technique. (B) Lacerated tailfin over time from a larva 3 days post fertilization (dpf) expressing LifeAct-EGFP in basal cells (*TgBAC(ΔNp63:Gal4); Tg(UAS: LifeAct-EGFP); Tg(hsp70:myl9-mApple))*. mpw: minutes post wounding. (B–F) are all maximum-intensity Z-projections of spinning-disk confocal images. (C) Individual cell from 3 dpf larva expressing LifeAct-EGFP mosaically in basal cells (*TgBAC(ΔNp63:Gal4)* larva injected with *UAS:LifeAct-EGFP* plasmid at the 1-cell stage). Wound was to the right approximately 1–2 min earlier. (D) Cells in a lacerated tailfin over time from 3 dpf larva expressing LifeAct-EGFP in basal cells (*TgBAC(ΔNp63:Gal4); Tg(UAS:LifeAct-EGFP); Tg(hsp70:myl9-mApple))*, approximately 1–2 min post wounding. Arrowheads: examples of individual actin-rich protrusions are followed over time. (E) Kymograph indicating the speed of basal cells at a given distance from the wound over time (N = 8 larvae). Line graphs show net displacement over space (*right*) and time (*bottom*) for each individual larva. See *Methods* and *Figure 1—figure supplement 1* for details of motion tracking analysis. (F) Lacerated tailfin from larva expressing GCaMP6f in basal cells (*TgBAC (ΔNp63:Gal4)* larva injected with *UAS:GCaMP6f-P2A-nls-dTomato* plasmid at the 1 cell stage). mpw: minutes post wounding. Due to the large dynamic range in GCaMP intensity, these images were gamma-corrected with a gamma of 0.5 for display purposes. (G) Kymograph of GCaMP6f intensity, normalized by the coexpressed nuclearly localized dTomato intensity, and relative to the normalized intensity pre-wounding ($F_0$) (N = 7 larvae). (H) Line graph of normalized profiles of the average speed and GCaMP intensity over time, averaged over 300 µm of tissue closest to the wound. To emphasize comparison of the temporal relationship, profiles are rescaled to lie between 0 and 1 (in arbitrary units).

The online version of this article includes the following video and figure supplement(s) for figure 1:

**Figure supplement 1.** Comparison of laceration and tail transection wounding techniques, and overview of the procedure for tissue motion analysis.

**Figure 1—video 1.** Comparison of wound response in lacerated (*top*) and transected (*bottom*) tailfins.

https://elifesciences.org/articles/62386#fig1video1

**Figure 1—video 2.** Dynamics of a single basal epidermal cell during the wound response.

https://elifesciences.org/articles/62386#fig1video2

**Figure 1—video 3.** Dynamics of wound response.

https://elifesciences.org/articles/62386#fig1video3

**Figure 1—video 4.** Calcium dynamics following wounding.

https://elifesciences.org/articles/62386#fig1video4

intensity is much faster than the change in cell speed (*Figure 1H*). This suggests that cell movement is not directly triggered or regulated by the increase in calcium, though calcium may indirectly promote wound-induced migration by functioning as a permissive cue.

Taken together, our observations of cell migration following laceration injury demonstrate a stronger migratory response in the first few minutes compared to tail transection, with overall tissue reorganization and calcium dynamics comparable to those induced by other wounding techniques. With the laceration method, we observed prominent actin-rich lamellipodia and waves of calcium and cell migration that propagated outward from the wound site at dramatically different rates.

## The wound response is sensitive to external sodium chloride, independent of osmolarity

Next, we sought to determine how different physical cues might initiate the wound response in our laceration injury model. Previous work had shown that the wound response in zebrafish epidermis was inhibited by isosmotic environments (*Gault et al., 2014*). We confirmed this result by immersing larvae in typical freshwater medium (E3, osmolarity ~12 mOsmol/l) supplemented with sodium chloride to a final osmolarity of ~270 mOsmol/l, within the range of typical zebrafish interstitial fluid osmolarity (*Gault et al., 2014*; *Kiener et al., 2008*). Larvae wounded in isosmotic sodium chloride had a markedly reduced wound response compared to larvae in E3 (hypotonic treatment), as measured by average basal cell speed over time (*Figure 2A*, compare red with black trace).

Since osmotic pressure is generated by any compounds with low membrane permeability ('osmolytes'), the osmotic surveillance model for wound detection predicts that wound response should depend only on the external concentration of osmolytes and not on their chemical identity. To test this prediction, we compared isosmotic sodium chloride treatment with isosmotic treatments of choline chloride, sodium gluconate, potassium chloride, or sorbitol. We found that, although all isosmotic treatments did reduce average cell speed, sodium chloride had a uniquely strong inhibitory effect (*Figure 2A*, compare red with other traces; *Figure 2—video 1*). In contrast, the degree to which all other osmolytes inhibited a wound response was remarkably consistent with each other (*Figure 2A*, compare all other traces).

To further quantify the multifaceted differences in average migratory cell displacement over time across conditions, we treated each time profile, with 30 samples over 15 min, as a data point in a 30-dimensional space (each timepoint a separate dimension), and used principal component analysis (PCA) to identify the major modes of variation among these profiles. We found that over 80% of all variation could be collapsed into two principal components, which roughly corresponded to the overall amplitude of the migratory response (72% of variation) and the timing of the peak of cell movement (9% of variation), respectively (*Figure 2—figure supplement 1A*). When the coefficients of each profile in these first two PCA modes were plotted, profiles from unwounded larvae and larvae wounded in hyposmotic medium were situated at extreme ends of the PCA space, profiles from sodium chloride clustered near the profiles from unwounded fish, and profiles from the other isosmotic treatments fell along the continuum between the sodium chloride and the hyposmotic profiles (*Figure 2—figure supplement 1D*). The difference in magnitudes of the first principal component between sodium chloride treatment and all other treatments was statistically significant (p<0.001, one-way fixed-effects Welch's ANOVA $F_{(6, 19)}=130.9$, with Games-Powell post-hoc tests), while the differences among the other isosmotic treatments were not statistically significant (*Figure 2B*). This quantification emphasized that the unique effect of isosmotic sodium chloride treatment on wound-induced cell migration depended on ionic chemical identity and was distinct from its osmotic effect.

Looking more closely at the dynamics of the cytoskeleton in the basal cell layer, we noticed that—with the exception of the immediate vicinity around the wound—there was a striking lack of cytoskeletal reorganization in response to wounding in isosmotic sodium chloride (*Figure 2C*; *Figure 2—video 1*). In contrast, in all other isosmotic treatment conditions we observed transient polarization of the actin cytoskeleton in cells from the basal layer (*Figure 2C*). We quantified this observation by measuring the relative LifeAct intensity change throughout the tailfin. Because cells moved and deformed to different extents during wound closure, we warped images to computationally decouple changes in cell shape from changes in actin intensity, overlaid the cell intensity distributions at each timepoint (see *Figure 2D*, *Figure 2—figure supplement 1E*, and Methods), and measured the relative changes in LifeAct-GFP fluorescence intensity pixel-by-pixel across the whole tailfin over time. We excluded the area approximately one cell diameter away from the wound,

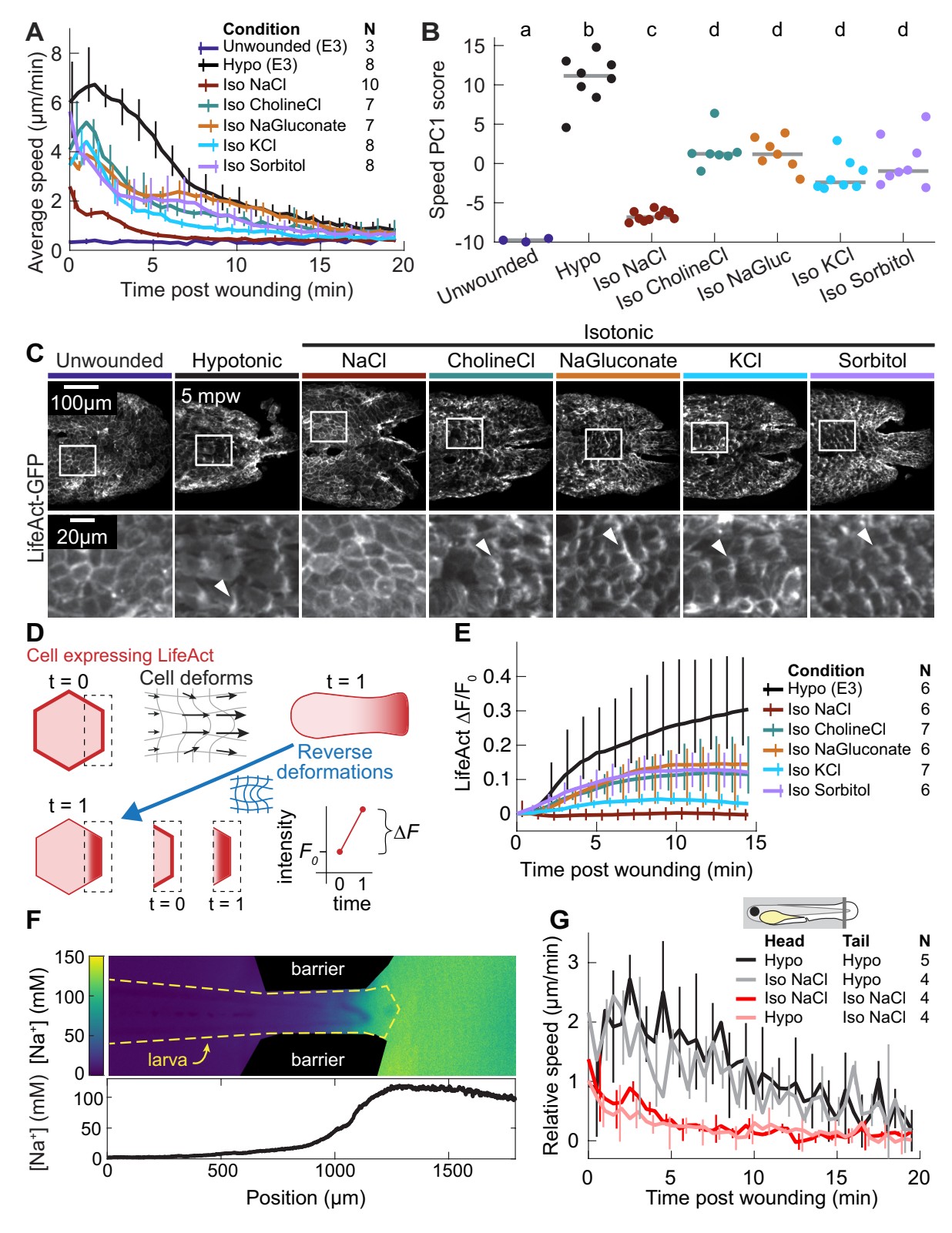

**Figure 2.** The wound response is sensitive to local concentrations of sodium chloride, independent of osmolarity. (**A**) Basal cell speed over time, averaged over 300 μm adjacent to the wound in each larva. 3 dpf larvae expressing LifeAct-EGFP in basal cells (*TgBAC(ΔNp63:Gal4); Tg(UAS:LifeAct-EGFP); Tg(hsp70:myl9-mApple)*) were incubated in E3 (Hypo) or E3 supplemented with 270 mOsmol/l of indicated osmolytes (Iso) and then the tailfin was lacerated and movement analyzed as described in *Methods* and *Figure 1—figure supplement 1*. N indicates the number of larvae in each

*Figure 2 continued on next page*

*Figure 2 continued*

condition. Error bars are bootstrapped 95% confidence intervals of the mean for each condition. (B) Speed trajectories for each larva were analyzed with PCA (see *Figure 2—figure supplement 1A–C*) and each trajectory's score along the first principal component is plotted. Gray bars indicate the mean PC1 score for that condition. Letters a-d indicate statistically distinguishable (significantly different) means (p<0.001, one-way fixed-effects Welch's ANOVA F(6, 19)=130.9, with Games-Howell post-hoc tests). See *Table 1* for p-values from post-hoc tests. (C) (*Top*) Representative tailfins from unwounded larvae or larvae wounded in different media. Images shown from 5 min post wounding. (*Bottom*) Insets shown below each image. Arrowheads: examples of polarized LifeAct intensity, in the direction of the wound. (D) Schematic of computational procedure for analyzing changes in intensity, after warping image to account for cell/tissue deformation. See *Methods* for more detail. (E) Relative pixel-wise change in LifeAct intensity over time, averaged over 300 μm adjacent to the wound in each larva. Error bars are bootstrapped 95% confidence intervals of the mean. (F) (*Top*) Image displaying the device allowing for different media compositions around the tailfin or the rest of the larva. Sodium concentration was calibrated with a sodium-sensitive fluorescent dye. (*Bottom*) Graph indicates the average sodium concentration along a line across the middle of the image. (G) Relative tissue speed for larvae with different media around their anterior or posterior, as shown in the diagram. To account for residual whole-larva movement due to peristaltic flow, the average tissue speed >300 μm away from the wound was subtracted from the average speed <300 μm away from the wound. Error bars are bootstrapped 95% confidence intervals of the mean.

The online version of this article includes the following video and figure supplement(s) for figure 2:

**Figure supplement 1.** Supporting information for *Figure 2*.

**Figure 2—video 1.** Wound response in different environmental conditions.

https://elifesciences.org/articles/62386#fig2video1

which was directly damaged by the laceration and would thus respond differently than cells further away. This analysis revealed that LifeAct intensity increased (often due to formation of lamellipodia) in all isosmotic treatments except sodium chloride, which did not induce any measurable cytoskeletal response at equivalent positions relative to the wound (*Figure 2E*). This indicates that sodium chloride's non-osmotic effect on the wound response is associated with inhibiting cytoskeletal polarization.

It is important to note that all of these experiments were done in the presence of the fish and amphibian anesthetic Tricaine, which is a voltage-gated sodium channel inhibitor and has been shown to inhibit tail regeneration in *Xenopus* tadpoles (*Ferreira et al., 2016*; *Tseng et al., 2010*). To rule out a potential effect of Tricaine on the wound response in isosmotic sodium chloride, larvae were immobilized by injection at the one-cell stage with mRNA encoding alpha-bungarotoxin, a component of the venom of the many-banded krait snake *Bungarus multicinctus*, which immobilizes by inhibiting nicotinic acetylcholine receptors at the neuromuscular junction (*Swinburne et al., 2015*). Larvae treated with this alternative inhibitor were not fully immobilized, presumably due to mRNA degradation at 3 dpf, but even so the wound response in larvae immobilized by alpha-bungarotoxin in isosmotic sodium chloride was nearly identical to the response of larvae treated with Tricaine (*Figure 2—figure supplement 1F*), suggesting the specific inhibitory effect of sodium chloride on the wound response is independent of the anesthetic used in the experiments.

Our findings are also robust to small variations in osmolarity: when solutions were deliberately prepared deviating from each other in osmolarity by 10% we found qualitatively similar responses in terms of migration and actin polarization (data not shown). This suggests that the unique effect of sodium chloride on the wound response is not due to small differences in osmotic strength between solutions with different ionic composition, but rather due to the actual chemical identities of the ions in solution.

## Wound response is determined by local wound environment

We next wished to determine whether the salt-specific role of sodium chloride in regulating actin reorganization during the wound response was due to local changes in sodium chloride in the wound vicinity or to global disruption of sodium chloride transport across the epidermis. Given the complex, ionocyte-mediated regulation of sodium chloride transport throughout the larval epidermis (*Guh et al., 2015*), it is possible that immersion of the entire fish in an isosmotic sodium chloride solution globally perturbs extracellular ionic composition throughout the larva. This pre-condition could lead to a general inhibition of a wound response, unrelated to location-specific cues that occur at the broken tissue barrier.

To distinguish between this global inhibition model and a model of local sodium chloride inhibition, we developed a two-chamber larval incubation device in which the tailfin was immersed in one

medium and the rest of the larva in another, with the distinct media compositions maintained by peristaltic flow (see *Methods* and *Figure 2—figure supplement 1G*). Control experiments using media with different sodium chloride concentrations and a sodium-sensitive fluorescent dye as a reporter of sodium concentration confirmed that a ~tenfold difference in sodium concentration could be maintained between the two chambers for many minutes (*Figure 2F*). When the same media was present in both chambers of the device, cell movement in response to a wound was similar to the uniform incubation conditions. When isosmotic sodium chloride media was present on only the tail-fin, the wound response was identical to when the entire larva was immersed in that media (*Figure 2G*, compare red and pink traces). Moreover, when hyposmotic media was present only on the tailfin, the wound response was similar to that observed with uniformly applied hyposmotic media (*Figure 2G*, compare black and gray traces), suggesting that only the local ionic environment regulates the wound response.

## Isosmotic solutions cause comparably low cell swelling regardless of composition

What is it about the chemical composition of these media that cause them to differentially induce actin polarization and cell movement? Although we ruled out differences in *osmolarity*, it is possible that these solutions differ in *tonicity* with respect to the basal cell membrane: if the cell membrane is differentially permeable to particular salts, identical concentrations of different salts may differentially induce water flow across the cell membrane. To determine whether differential swelling mediates the injury signal in different isosmotic environments, we directly measured the volume of cell clusters in each condition by mosaically expressing cytoplasmic mNeonGreen in basal cells within ~250 μm anterior of the tailfin. To measure volume, we obtained the projected area of the cluster and calculated the height at each pixel, and then integrated under this 'height map' to obtain an estimate of total cell volume.

We found that basal cells near the wound swelled dramatically within 90 s after wounding in hyposmotic media (*Figure 3A*). To facilitate comparison across cell clusters of varying sizes, we normalized the cluster volume to the volume prior to wounding and observed the relative change in volume over time (*Figure 3B*). This revealed that basal cell clusters from larvae wounded in hyposmotic media swelled by 50% of their initial volume on average and gradually shrank over 15 min. In contrast, basal cell clusters from larvae wounded in isosmotic media of any composition increased in volume only slightly, with cells in sodium gluconate swelling the most—an increase of less than 6% on average.

We used a paired data estimation plot (*Ho et al., 2019*) to visualize the absolute change in volume from before wounding to 90 s post-wounding in different media conditions (*Figure 3C*). For isosmotic media containing sodium chloride or choline chloride, the cellular volume change over this time frame was not statistically significant, nor was the magnitude of volume change in these media statistically distinguishable from that for cells on unwounded fish (p<0.05, two-sided t-tests on the paired average volume difference from each larva). While the increase in volume in both hyposmotic medium and sodium gluconate was statistically significant, the effect size in sodium gluconate was small: the mean paired volume increase between pre- and post-wounding for clusters in sodium gluconate was 0.07 picoliters (pl), while the mean paired volume increase for clusters in hyposmotic media was about 0.40 pl (95% C.I. 0.29–0.59 pl).

To test whether such slight swelling in isosmotic media other than sodium chloride was sufficient to explain the dramatic increase in actin polarization and migration in those media, we induced a limited degree of swelling in an orthogonal manner, by wounding larvae in intermediate concentrations of sodium chloride, and measured the degree of cell swelling and migration. As the concentration of sodium chloride decreased from isosmotic, we observed more cell migration (*Figure 3D*), but also more swelling immediately after wounding (*Figure 3E*). A linear relationship ($r^2$ = 0.95) was observed between initial volume change and degree of cell migration for the four conditions in which the concentration of sodium chloride was varied (*Figure 3F*), while the conditions in which different salts were used did not follow this same linear relationship. Instead, cells exposed to isosmotic salts other than sodium chloride moved substantially more than would be expected based solely on their volume change.

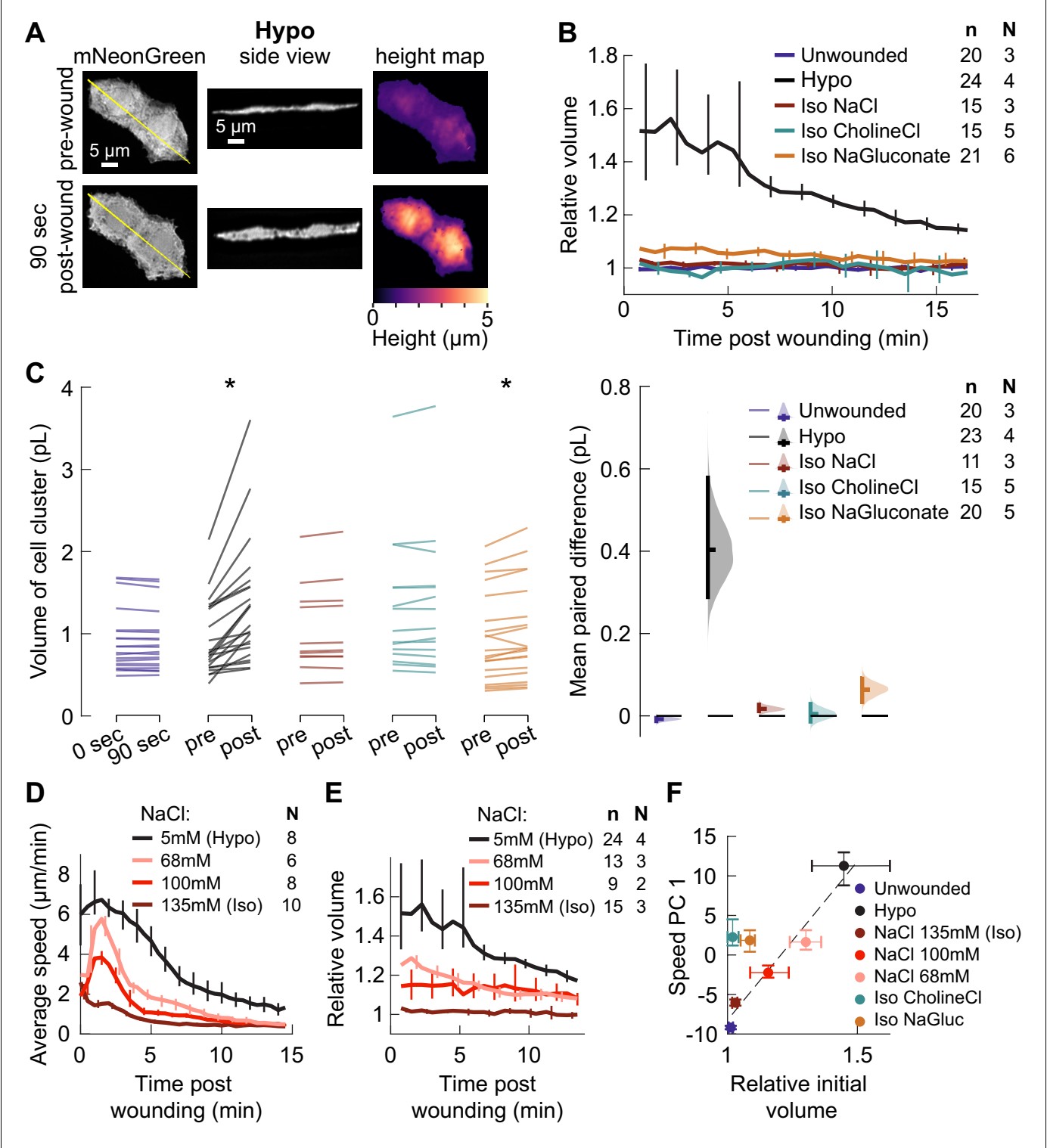

**Figure 3.** Isosmotic solutions cause comparably little cell swelling regardless of composition. (A) Overview of volume measurement for cell clusters. Representative cluster of cells from three dpf larvae mosaically expressing cytoplasmic mNeonGreen in basal cells (*TgBAC(ΔNp63:Gal4)* embryos injected with *UAS:mNeonGreen-P2A-mRuby3-CAAX* plasmid at the 1 cell stage). (*Left*) Z-projection of a representative cell cluster before and 90 s after wounding. (*Center*) side view at the position indicated by the yellow line. (*Right*) cell height measured at each pixel. (B) Average volume over time for cell clusters exposed to different media, relative to their volume before wounding. n: number of cell clusters. N: number of larvae. Error bars are 95% bootstrapped confidence intervals of the mean. (C) (*Left*) Absolute volume measurements for different cell clusters before and 90 s after wounding. *p<0.05, two-tailed t-test on the average paired difference from each larva. See *Table 2* for exact p-values and Cohen's d. (*Right*) mean paired

*Figure 3 continued on next page*

*Figure 3 continued*

difference of the measurements shown at left. Distributions are kernel density estimates. (D–E) Average tissue speed (D) and cell cluster volume (E) over time for larvae treated with different concentrations of sodium chloride. 5 and 135 mM speed data are the same as in *Figure 2A*; 5 and 135 mM volume data are the same as in (B). (F) Average cell cluster volume 90 s after wounding relative to pre-wounding volume, plotted against the average 1<sup>st</sup> principal component score for speed trajectories. Error bars are 95% bootstrapped confidence intervals of the mean. Linear regression of the Hypo and three NaCl conditions displayed with a dashed line ($r^2$ = 0.95).

## Electric fields are sufficient to induce cell migration in the absence of wound stimuli

Having ruled out differential swelling as the cause for the specific effect of sodium chloride on the injury response, we next turned to other aspects of fish physiology that are specifically affected by sodium and chloride ions and that could lead to a differential wound response in isosmotic external concentrations of sodium chloride. One such physiological cue is the lateral electric fields generated during injury by disruption of the TEP—which itself is generated by the transport of sodium and chloride ions across the skin (*Figure 4A*; *Dietz et al., 1967*; *McCaig et al., 2005*; *McCaig and Robinson, 1982*; *Potts, 1984*; *Reid et al., 2005*).

To test whether the basal layer of the epidermis will respond to electric fields in vivo in the absence of other wound cues, we constructed an apparatus with which to electrically stimulate the larval epidermis, simultaneously measuring the current flowing through the epidermis and visualizing the response of basal cells with microscopy (*Figure 4B*; *Figure 4—figure supplement 1A*). To apply an electric field in the skin, we used glass microelectrodes with a combined resistance of 18.5 ± 1.0 MΩ (n = 3, s.e.m.) when filled and immersed in a solution of 135 mM NaCl (chosen to approximate the composition of interstitial fluid). Silver chloride wires inserted into the microelectrodes provided the connection to electrical equipment.

Keeping in mind that many cells appear to respond to current density rather than electric fields per se (the two are proportional in a medium of a given ionic composition), we aimed to match the current density we applied to the lowest current density that induces a maximal directional response in cultured fish basal epidermal cells—approximately 500 mA/cm$^2$ (*Allen et al., 2013*). A difficulty in comparing our measurements to the applied currents in cell culture experiments is that the cross-sectional area through which the electric field propagates in the larva is not known. An upper bound on this area is the cross-section of the entire larva, which we estimate as approximately $10^{-4}$ cm$^2$ (*Figure 4—figure supplement 1B*). However, if current is only flowing through narrow interstitial spaces between epidermal cells, the cross-sectional area could be several orders of magnitude smaller than this, which would yield a larger current density for the same applied current.

Applying 1 µA of current, and assuming current passes through the total larval cross-section, we obtain a lower bound for the current density of 10 mA/cm$^2$, within the range of the reported measurements of cell sensitivity in cell culture. If the current was restricted to flow through a 0.5 µm-thick region around the edge of the tissue, such as through interstitial spaces within the epidermis, the effective cross-sectional area would decrease by a factor of 50, and the estimated current density would instead be 500 mA/cm$^2$, still within the range of sensitivity observed in culture (*Figure 4—figure supplement 1B*). Thus, we believe that a target current of 1 µA is within a physiological range for cell sensitivity.

**Table 2.** p-values for the two-tailed *t*-test on the paired differences in volume before and after wounding (*Figure 3C*), rounded to two significant digits, and Cohen's *d*, defined as the mean paired difference divided by the standard deviation of the differences.

| Condition | p-value | d |
| --- | --- | --- |
| Unwounded | 0.30 | –0.801 |
| Hypo | 0.021 | 2.24 |
| Iso NaCl | 0.20 | 1.07 |
| Iso CholineCl | 0.23 | 0.628 |
| Iso NaGluconate | 0.0079 | 2.20 |

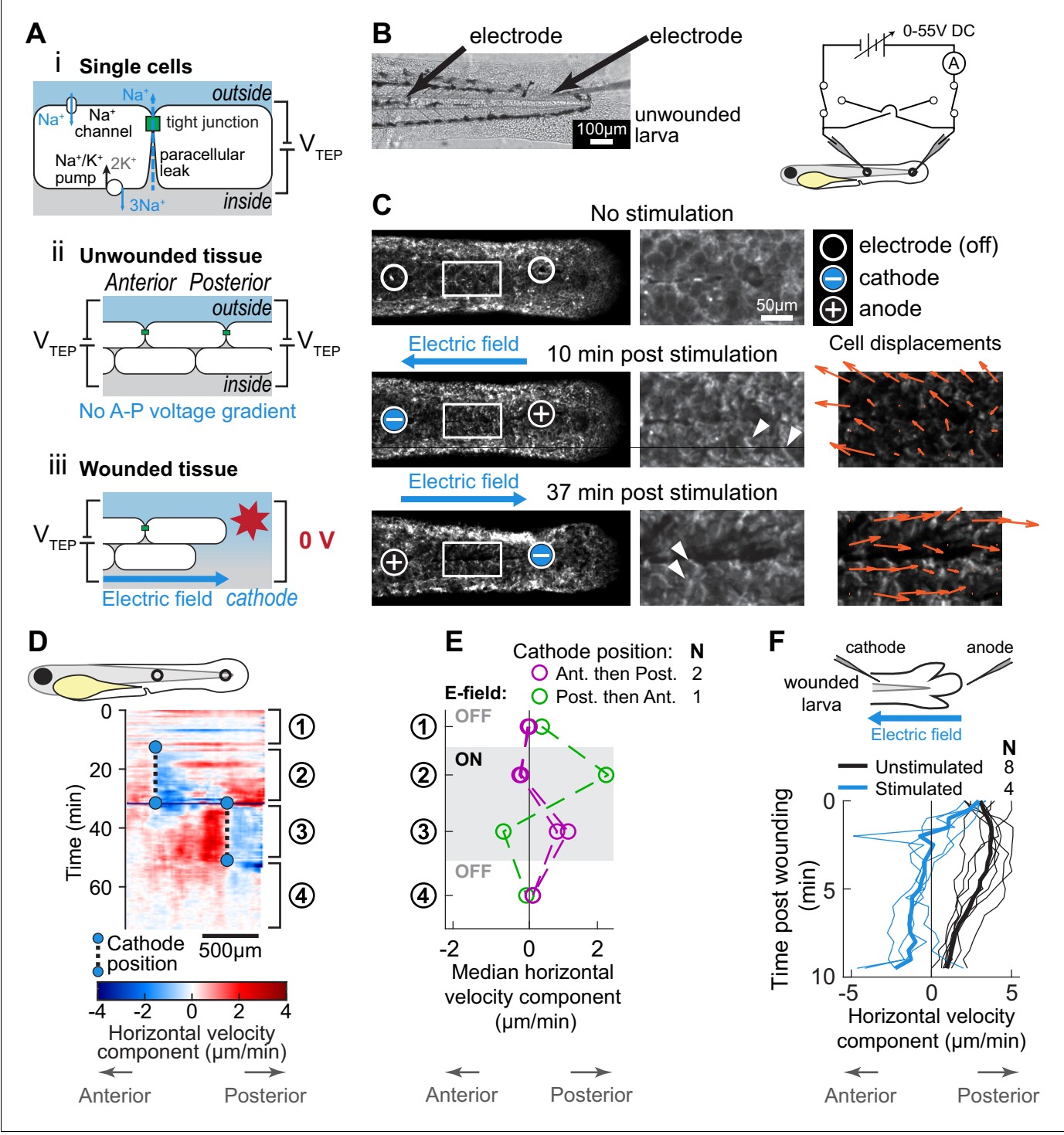

**Figure 4.** Electric fields are sufficient to induce cell migration in the absence of wound stimuli and can override endogenous wound signals. (**A**) (*i*) Schematic of the origin of the transepithelial potential (TEP) due to circulating flow of sodium ions. Other ions (such as chloride) may also be transported using energy derived from the sodium-potassium pump, and are not shown here for clarity. (*ii*) Unwounded tissue does not show an anterior-posterior TEP gradient. (*iii*) Wounding short-circuits TEP leading to an anterior-posterior TEP gradient and electric field. (**B**) (*Left*) Brightfield image of tailfin from 3 dpf larva expressing LifeAct-EGFP in basal cells (*TgBAC(ΔNp63:Gal4); Tg(UAS:LifeAct-EGFP)*), with electrodes inserted under the skin. (*Right*) Electrical stimulation circuit with variable DC voltage, current measurement, and switches to reverse current polarity in the larva. (**C**) Z-projections of LifeAct signal from larva shown in (**B**). (*Top*) electric field off; (*Middle*) electric field on with cathode at anterior electrode; (*Bottom*) electric field reversed with cathode at posterior electrode. Stills are from one continuous timelapse. Insets are shown, and displacement vectors from tissue motion tracking are shown in orange. Arrowheads: examples of polarized LifeAct intensity oriented toward the cathode. (**D**) Velocity kymograph

*Figure 4 continued on next page*

*Figure 4 continued*

from a representative timelapse. Color indicates horizontal velocity component from tissue motion tracking analysis. Blue circles and dashed lines indicate the position of the cathode when the electric field was turned on, roughly corresponding to the empty circles on the larva diagram. Numbers 1–4 indicate different phases of the timelapse. 1: electric field off; 2: electric field on, cathode anterior; 3: electric field on, cathode posterior; 4: electric field off. (E) Median horizontal velocity component from three different larva. Tissue velocity was averaged in the region between the two electrodes and then the median velocity during each phase 1–4 (described above) was plotted. For one larva (shown in green) the cathode was initially positioned at the posterior electrode and was then switched to the anterior electrode. (F) Average horizontal velocity component from stimulated or unstimulated larvae. In the stimulated condition, 3 dpf larvae expressing LifeAct-EGFP in basal cells (*TgBAC(ΔNp63:Gal4); Tg(UAS:LifeAct-EGFP); Tg(hsp70:myl9-mApple)*) were impaled with one electrode, with the other electrode positioned immediately posterior to the tailfin. Larvae were then wounded and the electric field turned on, with the cathode positioned at the anterior electrode. Tissue motion between the cathode and the wound was analyzed. Thin lines represent velocity for each larva. Thick lines represent average over larvae. Unstimulated data is the same as in *Figure 2A*.

The online version of this article includes the following video and figure supplement(s) for figure 4:

**Figure supplement 1.** Electrical analysis of the stimulation setup.
**Figure 4—video 1.** Effect of applied electric fields on cell migration in vivo.
https://elifesciences.org/articles/62386#fig4video1

We immersed larvae in hyposmotic medium and then impaled them with two glass microelectrodes connected in series to a variable DC power supply (*Figure 4B*). After waiting 10 min for the minor wound response from impalement to subside, we stimulated the larvae with a DC electric field, manually maintaining a current of approximately 1 µA (see *Methods*). We observed that cells situated between the two electrodes rapidly polarized their actin cytoskeletons and migrated in the direction of the cathode, changing direction when the polarity of the electric field was reversed (*Figure 4C–D*; *Figure 4—video 1*). The directed movement of cells toward the cathode persisted regardless of whether the cathode was initially at the anterior or posterior electrode (*Figure 4E*). This suggests that, within the native tissue context, electric fields are sufficient to induce and guide cell migration.

We next tested whether cues from exogenous electric fields could dominate over endogenous wound cues in vivo. We impaled anesthetized larvae with a single electrode, with the second electrode placed in the media posterior to the tailfin. Immediately after lacerating the tailfin, we turned on the electric field, with the cathode at the anterior electrode, so that the resulting electric field was the opposite polarity as compared to the field expected to be induced by injury. In all larvae tested, cells were dramatically slower in the presence of the exogenous electric field, and some cells even moved away from the wound toward the inserted cathode, which was never observed in unstimulated larvae (*Figure 4F*). This demonstrates that, in addition to guiding cells in the absence of wound stimuli, exogenous electric fields are sufficient to override the wound response of skin cells in a living animal.

## Discussion

Our results suggest that there are at least two distinct ways in which epidermal cells detect tissue injury through changes in their external ionic environment. First, mixing of interstitial fluid with dilute external media causes cells to swell, which prompts a migratory response, as has been previously shown (*Enyedi et al., 2016*; *Gault et al., 2014*). Second, this fluid mixing specifically reduces the concentration of sodium and chloride ions around cells, which we demonstrated is sufficient to prompt actin polarization in basal epidermal cells near wounds, independent of cell swelling or any change in environmental osmolarity.

One mechanism for cells to detect wounds that would be consistent with this osmolarity-independent effect of sodium chloride is through direct detection of electric fields arising from ion transport across the epidermis. Epithelial ion transport is critical for aquatic species, which typically live in aqueous solutions of vastly different ion concentrations than their internal interstitial fluid. Sodium and chloride are the predominant ions in both the interstitial fluids and external environments of freshwater fishes, and these ions must be actively and continually absorbed from the environment to counteract leakage through tight junctions and urine (*Kirschner, 2004*; *Potts, 1984*). While transport mechanisms for these ions vary across species and environmental conditions, a consistent theme is that transport of sodium and chloride ions may be partially interdependent but are

ultimately distinct: for example, the sodium-potassium ATPase can pump sodium ions against their concentration gradients, while chloride ions can be exchanged for bicarbonate ions, the concentration of which is regulated by the actions of carbonic anhydrase and V-ATPase proton pumps that expel excess protons (*Guh et al., 2015*; *Kirschner, 2004*). The activity of these various pumps and transporters can lead to charge separation across the epithelium, which manifests as the TEP.

TEPs have been measured in a variety of freshwater fish species, including trout, goldfish, and killifish, as well as in freshwater invertebrates like crayfish (*Eddy, 1975*; *Kerstetter et al., 1970*; *McWilliams and Potts, 1978*; *Wood and Grosell, 2008*; *Zare and Greenaway, 1998*). Sodium and chloride transport will be influenced by both concentration gradients and the TEP, and because these are the predominant ionic species, their transport will in turn affect the steady-state value of the TEP. For example, it has been proposed that differential permeability of fish skin to sodium versus chloride would lead to differing transport rates of these ions across the skin, resulting in a so-called 'diffusion potential,' which could alter the TEP depending on the concentration of sodium and chloride in the external medium (*Eddy, 1975*; *McWilliams and Potts, 1978*; *Potts, 1984*). Such a mechanism suggests that sodium and chloride transport would have a particularly strong influence on the TEP. And as shown in *Figure 4A*, when the skin is breached, the established TEP will be short-circuited, leading to electrical potential gradients within the skin, which generate electric fields whose orientation will depend on the relative position of the wound (*Reid and Zhao, 2014*). Supporting this theoretical rationale for the specific relationship between sodium and chloride transport and the TEP in fish epidermis, the transport of these two ions has been empirically shown to be essential for TEP maintenance other tissues, including rat cornea and amphibian epidermis (*Dietz et al., 1967*; *McCaig et al., 2005*; *Reid et al., 2005*).

In light of this plausible connection between electrical signaling and sodium chloride ion transport in fish skin, we have shown that an electric field is sufficient to direct cell movement in vivo in the absence of other wound cues, and can even override endogenous wound cues at the same timescale as the normal wound response. Complementary to our electrical perturbation experiments in vivo, electric currents have been directly measured in the external environment just outside of a wound in several model systems, including ex vivo wounded rat cornea, newt and tadpole tail amputation, and even wounds in human skin (*Ferreira et al., 2016*; *Nawata, 2001*; *Reid et al., 2009*; *Reid et al., 2007*; *Reid et al., 2005*). These measurements have all been conducted within a few hours of wounding, which yields much larger current measurements than the small (but measurable) currents that emanate from a wound during regeneration in hours and days following injury (*Ferreira et al., 2016*; *Rajnicek et al., 1988*). Paradoxically, these early wound currents fall in the range of 5–500 $\mu A/cm^2$, three orders of magnitude lower than the applied currents that cultured cells can respond to *Allen et al., 2013*; *Cohen et al., 2014*; *Sun et al., 2011*. We believe that the most parsimonious explanation for this discrepancy is that the physiological currents to which cells respond in vivo are restricted to small intercellular spaces, so the effective cross-sectional area through which current passes is small, and the current density within those spaces is large (*Robinson and Messerli, 2003*). When these currents leave the tissue, they spread over a much larger area, and so the current density measured just outside of a wound will be much smaller than the current density inside the tissue, which cells are actually sensing.

How might an endogenous electric field be generated in the zebrafish skin? The complexity of ion transport in freshwater fish species—which have evolved elaborate ways to absorb ions from extremely dilute surroundings—poses a challenge to unraveling the molecular mechanism underlying the electrical activity of the zebrafish epidermis. At least five types of ionocytes are responsible for maintaining ion homeostasis in zebrafish, each with a distinct complement of ion pumps, channels, and transporters, coupled by distinct electrochemical gradients (*Guh et al., 2015*). The contributions of each of these cell types to electrical activity of the skin has not been explored. Furthermore, traditional pharmacological approaches to dissecting ion transport mechanisms have had mixed success in zebrafish; for example, teleosts do not have a homolog to the amiloride-sensitive sodium channel implicated in establishing the TEP in frog skin, and the channel responsible for sodium uptake from the external environment remains unknown (*Venkatesh et al., 2007*). Given the genetic tools available in zebrafish, we believe these problems are surmountable, and deserving of further study given our results suggesting a connection between wound healing and epithelial electrophysiology.

At the cellular level, how might epidermal cells sense electric fields? Our results argue against a model in which this is accomplished primarily through changes in transmembrane potential:

treatment of larvae with isotonic KCl elicited a similar wound response to other non-NaCl osmolytes, despite the strong effect KCl generally has on transmembrane potential (*Figure 2A*). In culture, basal epidermal skin cells from fish respond to electric fields through an electrophoretic mechanism, in which an unidentified membrane-bound protein (or proteins) is dragged by electrophoresis to one side of the cell, promoting cytoskeletal polarization and initiation or redirection of motility through a primary PI3K-dependent pathway and secondary myosin II-dependent pathway (*Allen et al., 2013*; *Sun et al., 2013*). We suspect a similar mechanism is at work for the same cell type in zebrafish in vivo, and the electrophoretic mechanism is compatible with a model in which electric fields flow around cells within the epidermis immediately after wounding. Alternative and additional mechanisms for electric field sensing have been proposed in other cell types, including electrically stimulated calcium influx. Although calcium dynamics are not required for sensing electric fields in cultured fish basal cells (*Allen et al., 2013*; *Huang et al., 2009*), it is possible this signaling mechanism may be more relevant within intact tissue in vivo, where large calcium transients are observed immediately following injury (*Figure 1G*).

Electric fields are an attractive physical cue for wound detection because they are intrinsically directional, providing a mechanism to rapidly coordinate cell migration toward a wound at the spatial scales of tissue. In contrast, changes in osmolarity and cell swelling do not encode directional information; these cues can initiate a wound response but must be coupled with an additional spatial cue in order to orient cell movement appropriately. Electric currents have been measured emanating from wounds in human skin, and human skin and nasal epithelium both maintain a TEP, suggesting that electric fields may be relevant for wound healing in humans (*Foulds and Barker, 1983*; *Knowles et al., 1981*; *Reid et al., 2007*). Electric stimulation has been explored numerous times in clinical treatments for wounds and ulcers, but the lack of a detailed understanding of the mechanism by which electric fields influence cell behavior has limited progress (*Gentzkow et al., 1991*; *Zhao et al., 2020*). Our direct observation of actin polarization in response to electric fields in vivo at rapid timescales of 5–10 min bridges the gap between detailed mechanistic studies in cell culture and functional studies in tissues, and suggests that zebrafish are an ideal model system to further interrogate how cells respond to electric fields in physiological contexts.

# Materials and methods

**Key resources table**

| Reagent type (species) or resource | Designation | Source or reference | Identifiers | Additional information |
|---|---|---|---|---|
| Strain, strain background (*Danio rerio*) | TAB5 | | | WT background strain |
| Genetic reagent (*D. rerio*) | TgBAC(ΔNp63:Gal4)[la213] ; Tg(UAS:LifeAct-EGFP)[mu271] | PMID:25589751 | ZFIN: ZDB-FISH-200109–15 | |
| Genetic reagent (*D. rerio*) | Tg(hsp70:myl9-mApple) | PMID:25918227 | | |
| Recombinant DNA reagent | UAS:GCaMP6f-P2A-nls-dTomato | This paper | | Zebrafish expression vector containing the coding sequence for GCaMP6f and nls-dTomato separated by a self-cleaving peptide sequence P2A. Generated using Gateway cloning with Tol2kit plasmid backbones and AAV-EF1a-DIO-GCaMP6f-P2A-nls-dTomato as a template [Addgene plasmid #51083]. |

*Continued on next page*

*Continued*

| Reagent type (species) or resource | Designation | Source or reference | Identifiers | Additional information |
|---|---|---|---|---|
| Recombinant DNA reagent | UAS:mNeonGreen-P2A-mRuby3-CAAX | This paper | | Zebrafish expression vector expressing cytoplasmic mNeonGreen and membrane localized mRuby3, separated by a P2A sequence. Generated with Gateway cloning with Tol2kit plasmid backbones. |
| Recombinant DNA reagent | pmtb-t7-alpha-bungarotoxin | PMID:26244658 | RRID:Addgene_69542 | Vector for in vitro transcription of alpha-bungarotoxin mRNA |
| Sequence-based reagent | Primers for plasmid construction | This paper | | See **Supplementary file 1** |
| Chemical compound, drug | CoroNa Green | Invitrogen | Invitrogen:C36675 | Fluorescent sodium indicator dye |
| Software, algorithm | Code used for image analysis | This paper (**Kennard and Theriot, 2020**) | | The MATLAB and python code used for data analysis can be accessed at GitLab: https://gitlab.com/theriot_lab/fish-wound-healing-nacl |
| Software, algorithm | MATLAB | https://www.mathworks.com/products/matlab.html | RRID:SCR_001622 | Version R2018b |
| Software, algorithm | Fiji | https://fiji.sc | RRID:SCR_002285 | version 1.53 c |
| Software, algorithm | Python | https://www.python.org | RRID:SCR_008394 | version 3.7.3 |
| Software, algorithm | numpy | https://pypi.org/project/numpy/ PMID:32939066 | RRID:SCR_008633 | version 1.17.2 |
| Software, algorithm | scikit-image | https://pypi.org/project/scikit-image/ PMID:25024921 | RRID:SCR_008633 | version 0.15.0 |
| Software, algorithm | tifffile | https://pypi.org/project/tifffile/ | | version 2018.11.28 |
| Software, algorithm | OpenCV Python bindings | https://pypi.org/project/opencv-python/ | | version 4.1.1.1, including non-free algorithms |
| Software, algorithm | PIMS | **Allan et al., 2015** | | version 0.4.1 |

## Zebrafish husbandry

Zebrafish (TAB5 background wildtype strain) were raised and embryos harvested according to standard procedures (**Westerfield and Zon, 2007**). Experiments were approved by University of Washington Institutional Animal Care and Use Committee (protocol 4427–01). Animals were reared on a 14 hr light, 10 hr dark cycle at 28.5 °C in 1 to 9 L polycarbonate tanks (Aquaneering). Animals were crossed through natural spawning, and embryos were collected within 1–2 hr after spawning and raised in 100 mm petri dishes with 30–50 other embryos. Embryos were reared at 28.5 °C in E3

medium without methylene blue (5 mM NaCl, 0.17 mM KCl, 0.33 mM $CaCl_2$, 0.33 mM $MgSO_4$) (*E3 medium, 2008*). All experiments were performed on larvae 72–90 hr post-fertilization.

## Transgenic zebrafish lines

The *TgBAC(ΔNp63:Gal4)[la213]*; *Tg(UAS:LifeAct-EGFP)[mu271]*; *Tg(hsp70:myl9-mApple)* line was generated from a natural cross of the *TgBAC(ΔNp63:Gal4)[la213]*; *Tg(UAS:LifeAct-EGFP)[mu271]* line—a generous gift from Alvaro Sagasti (*Helker et al., 2013*; *Rasmussen et al., 2015*)—with the *Tg(hsp70:myl9-mApple)* line (*Lou et al., 2015*) by screening for fluorescence, and subsequently maintained through outcrosses to TAB5 WT fish.

## Plasmid constructs and mRNA synthesis

Plasmids for microinjection were generated using Gateway cloning into Tol2kit zebrafish expression vectors (*Kwan et al., 2007*). The *UAS:GCaMP6f-P2A-nls-dTomato* plasmid was generated by PCR amplification of a 2 kb fragment from *AAV-EF1a-DIO-GCaMP6f-P2A-nls-dTomato*, a gift from Jonathan Ting (Addgene plasmid #51083), using primers 1 and 2 (*Supplementary file 1*). This fragment was introduced into the Tol2kit plasmid pME (also known as pDONR221) using BP Clonase II and standard Gateway cloning procedures (Invitrogen). This pME plasmid was recombined with the Tol2kit plasmids p5E-UAS and p3E-polyA into Tol2kit expression vector pDestTol2CG2 to generate the final plasmid. All Tol2kit plasmids were a gift from C.-B. Chien.

To construct the *UAS:mNeonGreen-P2A-mRuby3-CAAX* plasmid, mNeonGreen (*Shaner et al., 2013*) was amplified from an encoding plasmid with primers 3 and 4, while mRuby3 (*Bajar et al., 2016*) was amplified from an encoding plasmid with primers 5 and 6 (*Supplementary file 1*). Following Gateway recombination into pDONR221 (mNeonGreen) and pDONR P2r-P3 (mRuby3) to create middle entry (ME) and 3'-entry (3E) vectors, respectively, Q5 mutagenesis (NEB) was used to introduce a P2A self-cleavage site to the C-terminal end of pME-mNeonGreen using primers 7 and 8, and a CAAX membrane localization tag was added to the C-terminal end of p3E-mRuby3 using primers 9 and 10 (*Supplementary file 1*). These modified plasmids were then recombined along with Tol2kit plasmid p5E-UAS into Tol2kit expression vector pDestTol2CG2 to generate the final plasmid. Plasmids containing the cDNA for mNeonGreen and mRuby3 were generous gifts from Darren Gilmour and Michael Lin, respectively. mRNA was synthesized using the SP6 mMESSAGE mMACHINE reverse transcription kit (Invitrogen). Alpha-bungarotoxin mRNA was synthesized from the plasmid *pmtb-t7-alpha-bungarotoxin*, a gift from Sean Megason (Addgene plasmid #69542) (*Swinburne et al., 2015*). Tol2 transposase mRNA was synthesized from the Tol2kit plasmid pCS2FA-transposase, a gift from C.-B. Chien.

## Microinjection

Embryos were injected at the 1- to 2-cell stage, into the cell (rather than the yolk). Plasmids were injected at a concentration of 20 ng/µl, with 40 ng/µl of Tol2 mRNA—the volume of these drops was not calibrated. For alpha-bungarotoxin mRNA injections, drops were calibrated to ~2.3 nl and 60 pg of mRNA was injected into each embryo.

## Preparation of larvae for imaging

Larvae were imaged at 3 days post-fertilization (3 dpf). One day prior to imaging, any larvae with the *hsp70:myl9-mApple* transgene were transferred from E3 at 28.5 ˚C into 20 ml scintillation vials of E3 pre-heated to 37 ˚C for 20 min before being returned to 28.5 ˚C.

Larvae were screened for transgenes of interest in the morning of 3 dpf. Larvae were anesthetized in E3 + 160 mg/l Tricaine (Sigma part number E10521) + 1.6 mM Tris, pH 7—hereafter referred to as E3 + Tricaine. Larvae were then mounted in 35 mm #1.5 glass-bottom dishes (CellVis D35-20-1.5N and D35C4-20-1.5N) with 1.2% low-melt agarose (Invitrogen) in E3 + Tricaine with the dorsal-ventral axis aligned parallel to the coverslip. Excess agarose was removed from around the tail of each larva with a #11 blade scalpel, and the incubation medium was replaced with experimental immersion medium prior to wounding and imaging.

E3 + Tricaine was the base for all experimental media. Additionally, isosmotic media was supplemented with 270 mOsmol/L of the indicated component.

## Tissue wounding

Solid borosilicate glass rods 1 mm in diameter (Sutter Instruments) were pulled into a needlepoint with a Brown-Flaming type micropipette puller (Sutter P-87). After the unwounded larva was imaged for several frames, timelapse acquisition was paused and the needle was maneuvered by hand to impale the larva at a position just dorsal (or ventral) to the posterior end of the notochord (see *Figure 1A*). The needle was then dragged posteriorly through the tailfin to tear the skin. This was repeated on the ventral (or dorsal) side of the notochord and then imaging was resumed. The entire procedure took 30 s – 1 min.

For tail transection wounds the procedure was very similar, except a #10 blade scalpel was manually maneuvered to cut off the tail posterior to the notochord, perpendicular to the anterior-posterior axis.

## Two-chamber device experiments

Two-chamber devices were made from polydimethylsiloxane (PDMS) cast in a mold fabricated from cut acrylic, inspired by previous work (*Donoughe et al., 2018*; *Huemer et al., 2017*). Device molds were cut from extruded acrylic (McMaster) using a Dremel LC-40 laser cutter and fused with acrylic cement. A 14 mm-long piece cut from the inner portion of a 22G spinal tap needle (Beckton Dickinson,~375 µm in diameter) was laid across the bottom of the mold to provide a channel for positioning the larva between the two chambers, and for fluid inlet into each chamber. A diagram of the device is shown in *Figure 2—figure supplement 1G*.

Prior to casting, the mold was pre-coated with 5% (w/v) Pluronic as a release agent. Sylgard 184 PDMS was mixed at a ratio of 10:1 (base: initiator), degassed, poured into the molds, degassed again, and polymerized at 50 ˚C overnight. PDMS devices were cleaned with dish soap, sonication, and Type I water, air plasma-treated for 1 min at 500 mTorr (Harrick Plasma PDC-001) and then immediately bonded to a #1.5 25 mm round glass coverslip.

Larvae were anesthetized in E3 + Tricaine and then immobilized within the device with 1.2% low-melt agarose. This agarose was carefully removed around the tail, keeping a plug of agarose around the larva in the anterior chamber for immobilization and to prevent convective fluid mixing. Tubing for peristaltic flow was positioned using custom-built equipment (*Figure 2—figure supplement 1G v-vii*). To maintain a stable concentration gradient, peristaltic flow was maintained in each chamber at a rate of approximately 0.3 ml/min.

To measure the concentration gradient that can be maintained in this device, solutions of E3 + Tricaine and E3 + Tricaine + 135 mM sodium chloride were prepared and supplemented with 10 µM of CoroNa Green (Invitrogen Cat#C36675). A wildtype larva was mounted in the device as described above and confocal image stacks were collected over time as the two CoroNa Green containing media were flowed into either chamber of the device. From the moment at which CoroNa Green was first detectable in the field of view, it took about 12 min for the concentrations in both chambers to stabilize. The fluorescent intensity was converted into a sodium concentration by first background subtracting and flat-field correcting each Z-projection (*Model and Blank, 2006*), and then comparing pixel intensities to a fluorescence standard curve generated by imaging drops of E3 + Tricaine + 10 µM CoroNa Green + sodium chloride (at different concentrations) and subjecting those images to the same intensity correction procedure. The standard curve was fit to a binding curve of the form $I = \frac{I_{max}[Na]}{K_d + [Na]}$ using nonlinear regression (fitnlm in MATLAB). The fit value of $I_{max}$ was 1.1529 (arbitrary units) and $K_d$ was 138 mM.

## Electrical stimulation

Microelectrodes were pulled from thin-walled borosilicate glass capillary tubes (1 mm O.D., 0.75 mm I.D., World Precision Instruments) with a Brown-Flaming type micropipette puller (Sutter P-87) and filled with 135 mM NaCl solution. The combined series resistance of both electrodes when immersed in the same solution was ~18.5 ± 1 MΩ (n = 3 independent electrode pairs, s.e.m.). Microelectrodes were connected into an electrical circuit using chlorided silver wires, with a variable DC power supply (PiezoDrive PD-200). Current was measured at 1 Hz sampling rate according to Ohm's Law, by recording the voltage drop across a 100 kΩ resistor in series with the larva using a multimeter (Fluke 287).

3 dpf larvae were mounted on 25 mm #1.5 coverslips in an open bath chamber (Warner RC-40LP) and immobilized in E3 + Tricaine containing 1.2% low-melt agarose (Invitrogen). Agarose was removed from above the larvae with a #11 scalpel, leaving a thin layer of agarose surrounding and immobilizing the larva. Electrodes were maneuvered into the larval trunk with micromanipulators (Narishige MN-153). After allowing the skin to recover from impalement for 10 min with the circuit thus connected through the larva, the power supply was turned on.

Voltage was manually varied to maintain a current of approximately 1 µA; to understand the magnitude of voltage required, we need to understand the three idealized sources of electrical resistance in the setup: the electrodes themselves ($R_{electrodes}$), the resistance of the larval tissue ($R_{larva}$), and the resistance of current that leaks out of the larva and flows through the medium instead, a so-called shunt pathway ($R_{shunt}$) (*Figure 4—figure supplement 1A*). The shunt resistance is difficult to measure, but will increase with decreasing medium conductivity (i.e. a more dilute solution would lead to a higher shunt resistance and more current flowing through the larva). In the following analysis, we assume that the shunt resistance is infinitely large; if instead the shunt resistance were finite, the magnitude of current passing through the larva would be lower than 1 µA.

The measurement of electrode resistance above (18.5 MΩ) is a lower bound on $R_{electrodes}$, since the conductivity of the 135 mM NaCl solution used for the measurement is high and no resistance arises due to liquid junction potentials, which emerge when electrode and bath solutions have different composition. Once the electrodes were inserted into the larva, the total resistance of the entire circuit was 30.9 ± 8 MΩ (n = 4, s.e.m.). Since the electrodes are being inserted into the interstitial space, in which the dominant ionic species is NaCl, we assume that this is a comparable setup to the reference measurement described above, and therefore infer that the $R_{larva}$ is approximately 11 MΩ, smaller than but the same order of magnitude as $R_{electrodes}$. The key upshot of this estimate is that a significant fraction of the applied voltage is actually dropping across the electrodes, and only about 35% of the applied voltage actually drops across the larva itself. In order to maintain a current of 1 µA, we applied voltages of 15–55V in our experiments, which translates to a potential of ~5–20V applied to the skin. The voltage increased over the course of the experiment, as resistance slowly but steadily increased.

For experiments combining electrical stimulation and wounding, one electrode was inserted into the larva and the other electrode was placed just outside and posterior to the larva. Tissue was lacerated as described above and the circuit was immediately turned on, with current manually maintained at approximately 1.5 µA.

## Microscopy and image acquisition

Images were acquired with one of two microscope setups. The first microscope used was a Leica DMI6000B inverted microscope equipped with a piezo-z stage (Ludl 96A600) a Yokogawa CSU-W1 spinning-disk confocal with Borealis attachment (Andor), a laser launch (Andor ILE) with 50 mW 488 nm and 50 mW 561 nm diode lasers (Coherent OBIS), 405/488/561/640/755 penta-band dichroic (Andor), and a 488/561 dual-band emission filter (Chroma ZET488/561m). A Plan Apo 20x NA 0.75 multi-immersion objective was used. On this Leica microscope, temperature was controlled with a closed forced-air temperature-controlled heating system to maintain temperature at 28–29˚C.

Alternatively, a Nikon Ti2 inverted microscope was used, equipped with a piezo-z stage (Applied Scientific Instruments PZ-2300-XY-FT), a Yokogawa CSU-W1 spinning-disk confocal with Borealis attachment (Andor), a laser launch (Vortran VersaLase) with 50 mW 488 nm and 50 mW 561 nm diode lasers (Vortran Stradus), 405/488/561/640/755 penta-band dichroic (Andor), and a 488/561 dual-band emission filter (Chroma ZET488/561 m). A Chroma 535/50 m emission filter was also used for volume measurements and electrical stimulation measurements, where only the green channel of emission light was collected. For standard wounding experiments, an Apo 20x NA 0.95 water immersion objective was used. For volume measurements, a Plan Apo 60x NA 1.27 water immersion objective was used. For electrical stimulation experiments without wounding, a Plan Fluor 10x NA 0.3 objective was used, while for electrical stimulation experiments with wounding the same 20x described above was used. On this Nikon microscope, larvae were maintained at 28–29˚C using a resistive heating stage insert (Warner DH-40iL) with a temperature controller (Warner CL-100).

On both systems, full-chip 16-bit 1024 × 1024 pixel images were acquired with a back-thinned EMCCD camera (Andor DU888 iXon Ultra) with Frame Transfer mode and EM Gain applied. Binning was not used, except for 2 × 2 binning for the GCaMP data in *Figure 1G*. MicroManager v1.4.23

(*Edelstein et al., 2010*) was used to control all equipment, including synchronizing rapid laser line switching and piezo-z positioning with camera exposures using TTL triggers.

Two-channel z-stacks were acquired at 30 s intervals, switching laser line at each z-position before changing z-position.

## Timelapse registration

To correct for whole-body movement and drift of the tailfin, registration was performed on movies from  *TgBAC(ΔNp63:Gal4)$^{la213}$*;  *Tg(UAS:LifeAct-EGFP)$^{mu271}$*;  *Tg(hsp70:myl9-mApple)* embryos. The myosin light chain-mApple was ubiquitously expressed, and we observed that only the skin cells migrated in response to wounding. We therefore considered myosin light chain fluorescence originating from tissues *other* than the skin to be stationary, and corrected any drift using this signal as follows. Prior to maximum-intensity projection, the LifeAct intensity was thresholded and used as a mask to set corresponding regions of the myosin z-stack to 0 using custom MATLAB code; following maximum-intensity z-projection, regions in the myosin channel that did not overlap with basal cells were emphasized. These modified z-projections of the myosin channel were manually cropped to select a 512 × 512 pixel region for registration >300 µm away from the wound. These subimages were registered in time with custom Python code by detecting KAZE features (*Lazebnik et al., 2012*), matching these features between adjacent timepoints, and fitting a Euclidean transform (rotation + translation) to the feature displacement vectors using RANSAC (*Fischler and Bolles, 1981*). The calculated transformations were then converted to the coordinates of the LifeAct image and used to register those z-projections. Registration was performed using custom Python scripts including the following libraries: numpy (*van der Walt et al., 2011*), scikit-image (*scikit-image contributors et al., 2014*), Tifffile (Christoph Gohlke, University of California, Irvine), and the python bindings for OpenCV (*Bradski and Dobbs, 2000*). Code is available on a GitLab repository (*Kennard and Theriot, 2020*; copy archived at swh:1:rev:67bba3afe283ece6e1e1c3db3b8234217ac5332c).

## Motion tracking and analysis

Registered LifeAct z-projections were manually aligned so the anterior-posterior axis was horizontal. Motion was tracked by detecting Shi-Tomasi corner points in each image (typically several thousand points per image) and tracking them from frame to frame using the Kanade-Lucas-Tomasi algorithm (*Lucas and Kanade, 1981*; *Shi and Tomasi, 1994*). These points correspond to areas of strong texture or curvature in the image, which make them straightforward to track. Due to high contrast and detail in the image, a majority of points could be tracked for the entire duration of a timelapse. Velocities could be calculated from the trajectories of these points.

The observation that most movement in the tailfin was in one primary direction (toward the wound) facilitated summarization of the data obtained from thousands of point tracks. To capture the most relevant tissue movement toward or away from the wound, the wounded region was manually traced, and the line between the centroid of all detected points and the centroid of the wound was computed. The positions of points (either the position in the first frame or the position in each frame) was projected onto this line to obtain a one-dimensional distance from the wound. Points were binned by their 1D coordinate along this line in 10 µm increments, and the average speed (in two dimensions) in each bin was calculated for each time point, providing a measure of velocity in one spatial dimension and time.

For the two-chamber device experiments (*Figure 2G*), drift of the tailfin due to peristaltic flow was not completely removed by the registration algorithm described above. To better compare measurements on different larvae, the relative velocity was used: the average velocity of points > 300 µm away from the wound centroid (along the line described above) was subtracted from the velocity of each point <300 µm away from the wound centroid.

For the electric field stimulation experiments without wounding (*Figure 4D–E*), the same motion tracking approach was used, but instead of averaging the speed (the magnitude of the 2D velocity of each point), the horizontal velocity component was averaged, so that positive and negative velocities indicated movement in the anterior or posterior direction, respectively.

## GCaMP intensity tracking

Maximum-intensity Z-projections were background-subtracted and manually rotated so the anterior-posterior axis was horizontal. The wound margin was manually traced, and the GCaMP6f and nls-dTomato intensities were each averaged in 10 µm increments based on the horizontal distance between each pixel and the wound centroid. To correct for variation in expression, the GCaMP6f intensity in each 10 µm increment was normalized to the nls-dTomato intensity in that increment, and then $F_t(x)$—the normalized GCaMP6f intensity at a horizontal position $x$ and frame $t$ —was further normalized to report relative changes in intensity over time, using the formula $\frac{\Delta F}{F_0} \equiv (F_t(x) - F_0(x))/F_0(x)$. This relative intensity in space and time was averaged over all fish to create a single intensity histogram.

## PCA of speed over time

Tissue speed within 300 µm of the wound centroid was averaged in each frame, and for each larva a track consisting of speed in the first 30 frames (15 min) was used for dimensionality reduction. The average speed for all 87 larvae over time was computed and subtracted from each track, and then PCA was performed on the 87 tracks in the 30-dimensional space.

## Non-rigid deformation of LifeAct distributions

Maximum-intensity z-projections of LifeAct in wounded tailfins were registered to remove rigid movement of the entire tissue as described above. A non-rigid warping was applied to further align individual cells, which moved at slightly different speeds in different directions. More explicitly, the goal was to identify warped coordinates $(\hat{x}, \hat{y})$, so that the fluorescence image $F_{t+1}(\hat{x}(x, y), \hat{y}(x, y))$ was aligned pixel-by-pixel to the previous frame, $F_t(x, y)$. To do this, the displacement field $D_t(x, y) = (\hat{x}, \hat{y})$ was computed, and the frame $t+1$ was warped using those coordinates, so that $F_{t+1}(D_t(x, y)) \sim F_t(x, y)$, where $\sim$ indicates similarity in intensity on a pixel-by-pixel basis. The displacement field was computed with the Diffeomorphic Demons algorithm (imregdemons in MATLAB with default settings) (**Vercauteren et al., 2009**). Displacement fields were iteratively composed to register the intensity in each frame to the first frame.

Once movies had been warped to align with the first frame, each frame was smoothed with a Gaussian filter and the relative change in fluorescence intensity at each pixel was computed according to $\Delta F/F_0 = (F_t(\hat{x}, \hat{y}) - F_0(x, y))/F_0(x, y)$. The average value of $\Delta F/F_0$ was calculated for each frame, excluding the region approximately one cell diameter away from the wound edge. Then these traces of $\Delta F/F_0$ were averaged across all larvae.

Upon inspection, some movies used for motion tracking analysis were not suitable for this analysis of change in LifeAct intensity, due to flickering of illumination light, which led to large frame-to-frame fluctuations in image brightness. Based solely on changes in the background intensity, the following criteria were used to exclude movies used in *Figure 2A* from analysis for *Figure 2E*:

1. The slope of a least-squares fit of average background intensity over time was greater than $0.005 \times m$ intensity units per frame, where $m$ is the median background intensity over the entire movie (typically around 500 intensity units).
2. The average background intensity in the first frame differed by more than $0.5 \times m$ from $m$, the median background intensity. (Because everything was normalized to the first frame, substantial deviations in the first frame affected the entire trajectory of $\Delta F/F_0$).

## Cell volume measurement

Z-stacks were acquired every 45 s at 60x magnification. Subimages of individual cell clusters were manually cropped and deconvolved using the Richardson-Lucy algorithm with 20 iterations in DeconvolutionLab2, a plugin for ImageJ (**Sage et al., 2017**; **Schindelin et al., 2012**). A maximum-intensity z-projection of a cell cluster was thresholded to obtain an x-y mask of the cell cluster. To obtain the height of the cell cluster at every other pixel in the x-y mask, the 3D image stack of a cell cluster was smoothed with a 3D gaussian filter and then edges were enhanced with a 3D Sobel filter. Then for a given pixel in the mask, the height was computed by first identifying two peaks in the linescan of fluorescence intensity along the z direction, and then computing the distance between the two peaks. For sub-pixel accuracy in z, the linescans were fitted to gaussians in the vicinity of the peaks.

To save on computation time, height was computed at every other pixel. Cell cluster height was spatially smoothed with a 2D median filter and then interpolated to generate a 'height map', the height of the cell cluster as a function of every pixel in the mask of the cluster. The volume of the cell cluster was obtained by numerically integrating this height map using the function integral2 in MATLAB.

The volume of each cell cluster over time was manually inspected for large discontinuities, and cell clusters for which the height maps had been obviously miscalculated—apparent by large frame-to-frame variations in cell volume over time, as well as visually obvious discontinuities in the height of the cell—were not included for further analysis.

## Statistical details

Each zebrafish larva was considered an independent biological replicate; there were no technical replicates in this work. When measurements were made on multiple cells in a single larva, those measurements were averaged to generate a single independent estimate per larva. Larvae were chosen for an experiment at random from a clutch of larvae.

For the ANOVA in *Figure 2B*, Welch's ANOVA with Games-Howell post-hoc tests was chosen due to unequal variances between conditions. The exact p-value for the Welch's ANOVA in *Figure 2B* was $2.9 \times 10^{-14}$. The p-values and effect sizes for the post-hoc Games-Howell tests are given in *Table 1*. The p-values and effect sizes for the paired *t*-tests in *Figure 3C* are given in *Table 2*.

## Acknowledgements

We thank Darren Gilmour and Jonas Hartmann for generously sharing their expertise, training, and advice on initially establishing this model system. We are also grateful to Jeff Rasmussen and Alvaro Sagasti for sharing of fish lines and expertise concerning zebrafish skin. We thank Philippe Mourrain and David Raible for graciously hosting fish in their facility and freely sharing reagents and advice, and thank Tom Daniel, Bill Moody, and Michael Kennedy for helpful discussions about epithelial electrophysiology. We are particularly grateful to Matthew Footer for invaluable technical advice, and also for careful critical reading of the manuscript, along with David Raible and Prathima Radhakrishnan. Finally, we are grateful to Ellen Labuz, Christopher Prinz, Mugdha Sathe, and other members of the Theriot laboratory, as well as Anna Huttenlocher, for numerous thought-provoking discussions. ASK was supported by NIGMS Training Grant T32GM008294; JAT acknowledges support from the Howard Hughes Medical Institute and the Washington Research Foundation.

**Table 1.** p-values (rounded to two significant digits) and Cohen's *d* for post-hoc Games-Powell statistical tests in *Figure 2B*. Cohen's *d* is defined as the difference in means divided by the square root of the average of the sample variances from the two samples being compared.

| | | Hypo | Iso NaCl | Iso CholineCl | Iso NaGluconate | Iso KCl | Iso Sorbitol |
|---|---|---|---|---|---|---|---|
| Unwounded | *p* | 0.0034 | 0.0027 | 0.0038 | 0.0038 | 0.0033 | 0.0035 |
| | *d* | −9.11 | −6.39 | −7.22 | −7.61 | −5.59 | −4.46 |
| Hypo | *p* | | 0.0033 | 0.0026 | 0.0026 | 0.0025 | 0.0023 |
| | *d* | | 7.57 | 3.29 | 3.54 | 4.49 | 3.28 |
| Iso NaCl | *p* | | | 0.0038 | 0.0037 | 0.0041 | 0.0068 |
| | *d* | | | −5.05 | −5.21 | −3.35 | −3.01 |
| Iso CholineCl | *p* | | | | 1.0 | 0.17 | 0.95 |
| | *d* | | | | 0.195 | 1.41 | 0.506 |
| Iso NaGluconate | *p* | | | | | 0.25 | 0.99 |
| | *d* | | | | | 1.280 | 0.364 |
| Iso KCl | *p* | | | | | | 0.86 |
| | *d* | | | | | | −0.622 |

## Additional information

### Funding

| Funder | Grant reference number | Author |
|---|---|---|
| Howard Hughes Medical Institute | | Julie A Theriot |
| Washington Research Foundation | | Julie A Theriot |
| National Institute of General Medical Sciences | T32GM008294 | Andrew S Kennard |

The funders had no role in study design, data collection and interpretation, or the decision to submit the work for publication.

### Author contributions

Andrew S Kennard, Conceptualization, Data curation, Software, Formal analysis, Validation, Investigation, Visualization, Methodology, Writing - original draft; Julie A Theriot, Conceptualization, Supervision, Funding acquisition, Project administration, Writing - review and editing

### Author ORCIDs

Andrew S Kennard (iD) https://orcid.org/0000-0002-0472-9144
Julie A Theriot (iD) https://orcid.org/0000-0002-2334-2535

### Ethics

Animal experimentation: Zebrafish (TAB5 background wildtype strain) were raised and embryos harvested according to standard procedures. Experiments were approved by the University of Washington Institutional Animal Care and Use Committee (protocol 4427-01).

### Decision letter and Author response

Decision letter https://doi.org/10.7554/eLife.62386.sa1
Author response https://doi.org/10.7554/eLife.62386.sa2

## Additional files

### Supplementary files

• Supplementary file 1. Primers used in Kennard and Theriot - Osmolarity-independent electrical cues guide rapid response to injury in zebrafish epidermis.

• Transparent reporting form

### Data availability

All data generated or analyzed during this study are included in the manuscript and supporting files. All code used to generate figures is available at https://gitlab.com/theriot_lab/fish-wound-healing-nacl (copy archived at https://archive.softwareheritage.org/swh:1:rev:67bba3afe283ece6e1e1c3db3-b8234217ac5332c/).

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
