## [Decision Letter]

**Acceptance summary:**

This paper examines the role of electrical fields in damage signalling for epithelial cell wounding using a zebrafish tail laceration model. While electrical fields had been previously noted in vitro, whether they played a role in early wound detection by epithelial cells has been unclear. The authors tracked the ability of epithelial cells to sense direction by imaging actin dynamics in zebrafish epidermis. From these studies, they find that directional sensing depends on the local concentration of specific electrolytes. Additionally, external electric fields can independently stimulate directional migration.

**Decision letter after peer review:**

Thank you for submitting your article "Osmolarity-independent electrical cues guide rapid response to injury in zebrafish epidermis" for consideration by *eLife*. Your article has been reviewed by three peer reviewers, and the evaluation has been overseen by a Reviewing Editor and Didier Stainier as the Senior Editor. The following individual involved in review of your submission has agreed to reveal their identity: Min Zhao (Reviewer #1).

The reviewers have discussed the reviews with one another and the Reviewing Editor has drafted this decision to help you prepare a revised submission.

Summary:

This paper examines the role of electric fields as a damage signal for epithelial cell wounding using a zebrafish tail laceration in vivo model. While electrical fields had been previously noted in vitro, whether they played a role in early wound detection by epithelial cells has been unclear. The authors tracked the ability of epithelial cells to sense direction by imaging actin dynamics in zebrafish epidermis. From these studies, they find that directional sensing depends on the local concentration of specific electrolytes. Additionally, external electric fields can independently stimulate directional migration.

Essential revisions:

Overall, the reviewers enjoyed the paper and found that this was an interesting system to analyze the role of electrical fields on wound healing and epithelial cell migration in general. They found the findings well supported and only felt cosmetic changes to the text were needed and a few extra bits of data included. These points were the main points that they wanted to see in the revision, however, I include all their comments for you below this:

– A more inclusive Introduction stating what was and was not known about electrical fields in wound healing previously.

– A discussion that touched on possible mechanisms for integrating electrical fields into directional migration. They realize that this is a big ask to address in experiments but it would be useful for further study to set up the narrative.

– Also, in the Discussion, it would be nice to discuss the magnitude of current and its relevance to physiological occurrence.

Reviewer #1:

This manuscript by Kennard and Theriot reports that electrical cues guide skin cells directional migration in response to injury. The authors bring molecular tools and analysis to study environmental cues, like osmolarity and electric fields in vivo. The effects of electrical cues are most studied in vitro. The in vivo model, the vivo approaches with molecular and imaging techniques bring bioelectricity research closer to mainstream techniques. Demonstrating of direct effect of electrical effects independent of osmolarity represent a significant step in this field. The results demonstrating the effects of NaCl, but not quite a few osmolarity control are impressive. I therefore believe the manuscript will be very interesting to *eLife's* readership.

I have the following questions and suggestions, which I do not expect the authors to address with new experiments, because as other pioneering research, this manuscript suggests more research questions/directions on the basis that it answers some very important questions. I believe perhaps the authors already have some results to some of those questions.

1) Good reason for choosing laceration over transection is given. I am a bit puzzled if the EFs and osmolarity are the mechanisms, why there were such difference? The endogenous EFs and osmolarity would be expected to be the same in both the laceration and transection models. Could the laceration stretch the tissue during injury procedure, so the marked increased migration was present in the laceration model? The stretch could activate stretch activated channels, stimulate cells, and realign matrix.

2) It is not clear what relationship can be established between GCaMP6f response and migration speed (Figure 1E, G, H). inhibition of the calcium response may help to test the relationship.

3) The local concentration of NaCl showed remarkable inhibitory effects on cell migration, and cell volume. As we know injury may activate channels and pumps, which then facilitate the ionic fluxes, thus generate persistent ionic currents. Channel and pump inhibition experiments could quickly point to some molecular basis of the involvement of NaCl.

4) I consider using Iso KCl is very interesting, because high K^+^ would significantly modulate cell membrane potential, however the effect on cell migration is very similar to those of Iso Choline Cl, iso NaGlunate, Iso Sorbitol. This would provide another side evidence for the role of wound electric fields in cell migration.

5) 200V DC is much higher than endogenous EFs expected in such a model. Caution should be given when interpreting the results. I also wonder whether the authors attempted experiments (Figure 4B, C) using wounded animals, perhaps the tissues after injury are not technically plausible (too fragmented) for such experiments.

6) One assumption in the paper is the TEP and wound EFs in vivo. Glass microelectrodes may be able to verify those in space and time. If this works (the TEP and wound EFs can be mapped), the effects of various treatments can be tested and exclude other possibilities.

Reviewer #2:

I enjoyed the manuscript. It is of general interest and would fit nicely in *eLife*. Driving cell movement and even overriding wound migrational cues with an electric field is very interesting. My principal concern is that it appears the manuscript has been written in a way to downplay the previous findings in this field. I am no expert on the effects of electric fields on wound healing and chemotaxis, but a cursory look at the literature shows that lot has been published in this arena. It appears that most if not all of the findings in this manuscript have been seen before in other contexts. I like the paper and feel it should be published even if none of the findings are novel. The zebrafish offers a great set of tools to interrogate electric fields on chemotaxis and wound healing. I am simply asking for a bit of clarity with respect to history of electrical fields cell chemotaxis and wound healing. The authors need to provide more context for their work in the Introduction with respect to electrical fields and more clearly describe what has been done before. In addition, the authors need to make additions to the conclusion that clearly define what is novel in their findings and how it relates to previous studies of electric fields and cell chemotaxis.

Reviewer #3:

This is an interesting paper examining the role of electric fields as a tissue damage signal for epithelial cells in vivo. Previous work had indicated the presence of electric fields in wounded tissues. But whether these phenomena play a role in early wound detection by epithelial cells has been unclear. The authors use live imaging in zebrafish to track the behaviour of epithelial cells in response to wounds. Imaging of actin dynamics was used as a readout for directional sensing in these cells. The authors show that directional sensing depends on the local concentration of specific electrolytes and that application of external electric fields can stimulate directional migration. These major conclusions are interesting and well supported. Although this is not the first time that electric fields are suggested to play a role, the study offers valuable direct evidence in vivo evidence, and introduces a new system in which the mechanisms can be studied further.

The study is focused on establishing whether electric fields play a role in wound sensing and does not touch on how these effects are mediated. The experiments were designed to distinguish osmotic from electric effects, establish whether the effects are global or local and assess the direct effects of electric fields on epithelial cell motion. These are significant and do not appear trivial. Nevertheless, some insight, even in the form of discussion, into how these effects might be sensed by epithelial cells seemed to be lacking. At the minimum, the authors could provide ideas based on the literature. Ideally, the study would include an analysis of cytoskeletal rearrangements and calcium dynamics in response to electric fields or alterations of electrolytes for completion. The authors introduce these key readouts of epithelial signalling, but they did not make full use of these in their functional assays. Depending on whether electric fields influence the calcium wave, different mechanistic hypotheses can be made for future studies.

---

## [Author Response]

Essential revisions:Overall, the reviewers enjoyed the paper and found that this was an interesting system to analyze the role of electrical fields on wound healing and epithelial cell migration in general. They found the findings well supported and only felt cosmetic changes to the text were needed and a few extra bits of data included. These points were the main points that they wanted to see in the revision, however, I include all their comments for you below this:– A more inclusive Introduction stating what was and was not known about electrical fields in wound healing previously.

We thank the reviewers for inviting a more comprehensive treatment of this very interesting literature. We have substantially expanded the section of the Introduction regarding previous work on electric fields, focusing on the most relevant work and highlighting open problems in the field. Upon reflection we believe that this addition substantially improves the clarity of the presentation for most readers.

– A discussion that touched on possible mechanisms for integrating electrical fields into directional migration. They realize that this is a big ask to address in experiments but it would be useful for further study to set up the narrative.

We have added new text in the Discussion speculating on the molecular mechanisms establishing electric fields in vivo (including some current challenges to experimentally identifying these mechanisms) as well as the mechanisms by which individual cells sense electric fields.

– Also, in the Discussion, it would be nice to discuss the magnitude of current and its relevance to physiological occurrence.

We have expanded our discussion of the magnitude of electric fields employed in this study in three parts of the manuscript: in the Results pertaining to Figure 4, in which we have included more details on our electrical setup and the choice of field strength, supported by a newly added figure supplement to Figure 4; in the Discussion, where we compare the magnitude of the applied fields in this work to the magnitude of endogenous wound currents reported in the literature; and in the Materials and methods, where we include some more technical analysis of the sources of resistance in our electrical setup.

Reviewer #1:[…] I have the following questions and suggestions, which I do not expect the authors to address with new experiments, because as other pioneering research, this manuscript suggests more research questions/directions on the basis that it answers some very important questions. I believe perhaps the authors already have some results to some of those questions.1) Good reason for choosing laceration over transection is given. I am a bit puzzled if the EFs and osmolarity are the mechanisms, why there were such difference? The endogenous EFs and osmolarity would be expected to be the same in both the laceration and transection models. Could the laceration stretch the tissue during injury procedure, so the marked increased migration was present in the laceration model? The stretch could activate stretch activated channels, stimulate cells, and realign matrix.

This is a good point, and the additional possibilities the reviewer raises are indeed worth mentioning. We have added text to the appropriate section of the Results mentioning these possibilities.

2) It is not clear what relationship can be established between GCaMP6f response and migration speed (Figure 1E, G, H). inhibition of the calcium response may help to test the relationship.

This is indeed an interesting experiment. An analogous experiment with nls-GCaMP6s expressed in all cells in 3 dpf zebrafish larvae has been performed in isotonic sodium chloride (Enyedi et al., 2016), showing that this particular ionic environment inhibited the calcium flickers occurring >1 min after wounding, but not the initial calcium burst propagating through the tissue. It is not known how a different osmolyte might affect these results.

While perturbing the calcium response would be useful for clarifying the role of calcium transients and cell migration, the primary focus of the calcium imaging in this work was to compare the laceration technique to other previously established work (e.g. Enyedi et al., 2016; Yoo et al., 2012). As such, we believe the inhibition of the calcium response would be quite interesting but beyond the scope of this paper.

3) The local concentration of NaCl showed remarkable inhibitory effects on cell migration, and cell volume. As we know injury may activate channels and pumps, which then facilitate the ionic fluxes, thus generate persistent ionic currents. Channel and pump inhibition experiments could quickly point to some molecular basis of the involvement of NaCl.

We appreciate the reviewer bringing up this issue. We have tried several of the obvious candidates—the sodium-potassium pump inhibitor ouabain, the V-ATPase pump inhibitor Bafilomycin A, and the sodium ionophore monensin—but our efforts were inconclusive. Teleosts do not have a homolog to the amiloride-sensitive sodium channel (Venkatesh et al., 2007), so this other obvious candidate inhibitor was a dead-end. For completeness we did try the amiloride analog benzamil, but did not observe any effect.

We were most interested in determining if we could inhibit wound healing in hypotonic solutions or relieve the inhibition of the wound response in Iso NaCl, but we observed neither effect with these drug treatments. Without a direct measurement of the transepithelial potential in untreated and drug-treated larvae, we believe it is difficult to rationalize any effect—or lack thereof—in the context of our model. We think this makes measuring the transepithelial potential the most critical next step in this research area (see our response to point 6 below). Another complicating factor is that the physiology of ion transport in freshwater fish skin and gills is complex and incompletely understood (see for example Guh et al., 2015). There are at least 5 different types of ion-transporting cells in the epidermis, each with their own repertoire of ion channels and pumps. We do not know how each of these cell types contributes to the TEP, how the activity of any of these cells might be altered by tissue injury, or if ion-transport by non-professional ion transporting cells may become relevant during wound healing. This complexity makes it non-trivial to design a targeted screen for drugs that might affect TEP in zebrafish, while an exhaustive screen was beyond the scope of this paper.

4) I consider using Iso KCl is very interesting, because high K^+^ would significantly modulate cell membrane potential, however the effect on cell migration is very similar to those of Iso Choline Cl, iso NaGlunate, Iso Sorbitol. This would provide another side evidence for the role of wound electric fields in cell migration.

We appreciate the interest in this experiment and we have emphasized the possible effect of KCl on membrane potential in the Discussion.

5) 200V DC is much higher than endogenous EFs expected in such a model. Caution should be given when interpreting the results. I also wonder whether the authors attempted experiments (Figure 4B, C) using wounded animals, perhaps the tissues after injury are not technically plausible (too fragmented) for such experiments.

We apologize for our lack of clarity on this point. The illustration in Figure 4 showed the maximum possible voltage of our power supply (200V), not the maximum voltage applied in any experiment—the latter is much more relevant. We have amended the figure to include the maximum voltage we used, which was 55V. However, we believe the most relevant metric for this cell type is not the electric field per se (in V/cm) but rather the current density experienced by cells within the tissue. We have added text in the Results and Materials and methods section in which we explicitly compare our measurements to the electrical sensitivities of stimulated cells in culture as well as new text to the Discussion comparing to the measurements of electrical currents emanating from wounds in vivo. In that text we argue for the physiological relevance of our electric perturbations.

We have indeed tried the experiment of applying electric fields to wounded animals, and by placing the cathode opposite the direction of the wound we could cause cells to slow their migration to the wound or even reverse direction. The results are in Figure 4F.

6) One assumption in the paper is the TEP and wound EFs in vivo. Glass microelectrodes may be able to verify those in space and time. If this works (the TEP and wound EFs can be mapped), the effects of various treatments can be tested and exclude other possibilities.

This is a fantastic idea. Inspired by work in *Xenopus* (Ferreira et al., 2016), we attempted to conduct similar measurements in the zebrafish epidermis, but were ultimately unsuccessful. Although we verified we could successfully place electrodes within the epidermis, we were unable to achieve consistent TEP readings from day to day.

One difficulty with these measurements compared to prior work is that zebrafish larvae are about half the size of *Xenopus* tadpoles at a similar stage (~3mm vs. 7mm in their longest dimension, and other dimensions are also proportionally smaller). Histology also suggests that there is very little space between the epidermis and other structures in zebrafish, while SEM of *Xenopus* larvae indicates an appreciable interstitial space below the epidermis into which an electrode may be reliably inserted without measuring other potentials (Hotary and Robinson, 1994). Another challenge is that ideally these experiments should be conducted in a hypotonic medium, to measure the TEP in as close to unperturbed conditions as possible, and determine how the TEP depends on the external ionic conditions. However, this presents unavoidable difficulties related to making electrophysiological measurements in dilute solutions. Notably, the liquid junction potential that arises from changing the local environment around the electrode tip from dilute freshwater to salty interstitial fluid is hard to correct for and can substantially bias the results, especially if the true TEP is a similar order of magnitude to the junction potential. We hope that our work inspires professional electrophysiologists to attempt these measurements in the future!

Reviewer #2:I enjoyed the manuscript. It is of general interest and would fit nicely in eLife. Driving cell movement and even overriding wound migrational cues with an electric field is very interesting. My principal concern is that it appears the manuscript has been written in a way to downplay the previous findings in this field. I am no expert on the effects of electric fields on wound healing and chemotaxis, but a cursory look at the literature shows that lot has been published in this arena. It appears that most if not all of the findings in this manuscript have been seen before in other contexts. I like the paper and feel it should be published even if none of the findings are novel. The zebrafish offers a great set of tools to interrogate electric fields on chemotaxis and wound healing. I am simply asking for a bit of clarity with respect to history of electrical fields cell chemotaxis and wound healing. The authors need to provide more context for their work in the Introduction with respect to electrical fields and more clearly describe what has been done before. In addition, the authors need to make additions to the conclusion that clearly define what is novel in their findings and how it relates to previous studies of electric fields and cell chemotaxis.

We thank the reviewer for the positive remarks, and appreciate the feedback on how we could improve our contextualization of our work. We have substantially revised the Introduction and Discussion to compare this work to other work in the field and how our work adds to the work that has already been done. We are encouraged that the reviewer agrees that establishing zebrafish as a promising model organism for bioelectric study is itself an important contribution, and we further argue that it is especially useful for probing a gap in the literature concerning the cytoskeletal response to electric fields in vivo, due to the relative ease of directly imaging changes in actin organization inside of epidermal cells at very high spatial resolution in these living animals.

Reviewer #3:[…] The study is focused on establishing whether electric fields play a role in wound sensing and does not touch on how these effects are mediated. The experiments were designed to distinguish osmotic from electric effects, establish whether the effects are global or local and assess the direct effects of electric fields on epithelial cell motion. These are significant and do not appear trivial. Nevertheless, some insight, even in the form of discussion, into how these effects might be sensed by epithelial cells seemed to be lacking. At the minimum, the authors could provide ideas based on the literature. Ideally, the study would include an analysis of cytoskeletal rearrangements and calcium dynamics in response to electric fields or alterations of electrolytes for completion. The authors introduce these key readouts of epithelial signalling, but they did not make full use of these in their functional assays. Depending on whether electric fields influence the calcium wave, different mechanistic hypotheses can be made for future studies.

We thank the reviewer for their feedback on our work. We have added new text to the Discussion connecting this work to previous studies of the mechanisms of electric field sensing.

We agree that an analysis of the cytoskeletal rearrangements during electric fields is a very promising avenue for future work in this system. We do demonstrate actin polarization in the presence of an electric field (Figure 4C), but our current data (taken at 10X magnification) is not of sufficient resolution to conduct a similar analysis of actin reorganization to that shown in Figure 2E. We hope that further data collected at higher magnification should be able to address this point.

The interplay between calcium and electric fields is also an interesting possibility. Calcium dynamics are not required for the response of fish basal epidermal cells to electric fields in culture, suggesting that the temporal coincidence between these two events in vivo may not be significant (Allen et al., 2013; Huang et al., 2009). We have included this information in the Discussion for further context.